# Retrv-R1: A Reasoning-Driven MLLM Framework for Universal and Efficient Multimodal Retrieval

**Lanyun Zhu**[1]   **Deyi Ji**[2]   **Tianrun Chen**[3]   **Haiyang Wu**[2]   **Shiqi Wang**[1†]

[1]City University of Hong Kong   [2]Tencent   [3]Zhejiang University

## Abstract

The success of DeepSeek-R1 demonstrates the immense potential of using reinforcement learning (RL) to enhance LLMs' reasoning capabilities. This paper introduces Retrv-R1, the first R1-style MLLM specifically designed for multimodal universal retrieval, achieving higher performance by employing step-by-step reasoning to produce more accurate retrieval results. We find that directly applying the methods of DeepSeek-R1 to retrieval tasks is not feasible, mainly due to (1) the high computational cost caused by the large token consumption required for multiple candidates with reasoning processes, and (2) the instability and suboptimal results when directly applying RL to train for retrieval tasks. To address these issues, Retrv-R1 introduces an information compression module with a details inspection mechanism, which enhances computational efficiency by reducing the number of tokens while ensuring that critical information for challenging candidates is preserved. Furthermore, a new training paradigm is proposed, including an activation stage using a retrieval-tailored synthetic CoT dataset for more effective optimization, followed by RL with a novel curriculum reward to improve both performance and efficiency. Incorporating these novel designs, Retrv-R1 achieves SOTA performance, high efficiency, and strong generalization ability, as demonstrated by experiments across multiple benchmarks and tasks. Project page.

## 1   Introduction

Information retrieval is a key direction in AI with broad applications, such as search engines [9, 5] and retrieval-augmented generation [28, 63]. Early work primarily focused on retrieval for single, fixed-format data types, such as text-to-text [76, 40], text-to-image [10, 73], and image-to-image retrieval [4, 51]. To improve efficiency and facilitate real-world deployment, recent efforts [59, 38, 34] have increasingly emphasized universal multimodal retrieval, where a single model is required to handle diverse retrieval tasks and data modalities simultaneously. Earlier work in this setting typically employs small foundation models such as CLIP, while more recent methods [38, 34, 77, 27] explore leveraging stronger Multimodal Large Language Models (MLLMs) to achieve further improvements, taking advantage of their rich pretrained knowledge and strong generalization capabilities.

Some retrieval methods [27, 77] apply MLLMs by leveraging their embeddings for similarity comparison, but their effectiveness and robustness are often unsatisfactory due to the imprecision of embeddings and similarity computations. Other approaches [34, 38] formulate retrieval as a question-answering (QA) task by constructing an input instruction from the query and candidates, prompting the MLLM to generate the retrieval result. These methods achieve improved performance by exploiting the strong QA capabilities of MLLMs. However, they typically rely on the MLLM to directly generate the result without any explicit reasoning process, which limits their ability to handle complex cases. Moreover, the supervised fine-tuning (SFT) strategies commonly used to train retrieval MLLMs

---

[†]Corresponding Author

39th Conference on Neural Information Processing Systems (NeurIPS 2025).

suffer from several shortcomings, such as poor generalization and frequent hallucinations, as widely reported in prior studies [68, 24, 7]. These challenges motivate us to explore a new paradigm for MLLM-based retrieval, one that incorporates stronger reasoning capabilities and a more effective training mechanism, thereby achieving higher retrieval accuracy and greater reliability.

Recent studies [61, 67, 43] have shown that reinforcement learning (RL) can effectively optimize a model's reasoning abilities, enabling LLMs to solve complex tasks, such as mathematics and programming, more effectively through step-by-step chain-of-thought (CoT) reasoning. As a prominent example of such RL-based methods, DeepSeek-R1 [14] has achieved particularly remarkable success, significantly improving LLM performance by utilizing a novel Group Relative Policy Optimization (GRPO) algorithm. Moreover, DeepSeek-R1 computes rewards using predefined rules, rather than relying on additional models as in many previous RL approaches, thereby simplifying the pipeline and reducing optimization complexity. These advantages inspire us to explore the question: *whether the paradigm of DeepSeek-R1 can also be leveraged to enhance the capabilities of retrieval MLLMs?*

Based on the above motivations and to answer the aforementioned question, in this work, we propose Retrv-R1, the first R1-like reasoning MLLM framework designed for universal multimodal retrieval tasks. As an initial exploration, we first attempt to directly train a standard MLLM using the naive GRPO on the retrieval datasets. However, the performance is unsatisfactory due to the following issues: (1) We observe that the model's convergence is difficult and unstable. Moreover, the trained model often generates incorrect reasoning processes, leading to suboptimal performance and inaccurate retrieval results. This aligns with observations from prior work [16], which indicates that directly applying RL struggles to effectively incentivize the reasoning capabilities of MLLMs, especially when the training data is limited in quantity or quality. (2) During inference, feeding tokens from all candidates into the MLLM introduces significant computational and memory overhead. This issue becomes particularly problematic when CoT reasoning is involved, as the fine-grained reasoning process consumes even more tokens, potentially exceeding the model's context length and available computational resources. To address these challenges, we introduce a novel information compression module alongside a details inspection mechanism. In this approach, most candidates are compressed into a small number of tokens, while only a few challenging candidates, identified automatically by the MLLM during the CoT process, retain their full features. This strategy not only effectively reduces the token count, thus preserving sufficient context for CoT reasoning, but also mitigates the negative impact of compression-caused information loss on the judgment of difficult candidates. Additionally, we propose an activation method prior to RL training, where a retrieval-tailored synthetic CoT dataset is generated and used for SFT, thereby avoiding the optimization difficulties arising from directly applying RL. We also introduce a novel curriculum-based mechanism for the RL reward, enabling the MLLM to develop strong reasoning abilities while maintaining high efficiency.

Incorporating these novel designs, Retrv-R1 constitutes a powerful and efficient retrieval MLLM framework. We conduct extensive experiments across multiple benchmarks, such as the universal retrieval benchmark M-BEIR and out-of-domain benchmarks on multimodal recommendation tasks. The strong performance across diverse settings demonstrates the high effectiveness and generalization ability of Retrv-R1. In summary, the main contributions of this work are: (1) We propose Retrv-R1, the first R1-style reasoning MLLM framework specifically designed for universal retrieval tasks. (2) We introduce several innovative and task-tailored designs within the Retrv-R1 framework, including a new model structure to reduce token consumption, a details inspection mechanism to address challenging candidates, and a curriculum-based reward system to improve both effectiveness and efficiency. (3) Extensive experiments across multiple benchmarks and task settings demonstrate the strong performance, efficiency, and generalization ability of Retrv-R1.

## 2 Related Work

**Multimodal Retrieval.** The development of deep learning [86, 81, 80, 84, 82, 18] has driven rapid progress across a wide range of retrieval tasks, such as text-image cross-modal retrieval [45, 10, 73, 8, 23, 22], composed image retrieval [4, 51, 13, 55, 3], multimodal document retrieval [6, 15, 39], and instruction-based image retrieval [62, 72, 1]. In recent years, vision-language models such as CLIP [48] have achieved remarkable success in this domain [4, 59, 50, 44, 21]. For example, [22] enhances CLIP through prompt tuning to enable highly generalizable retrieval. More recently, MLLMs have been applied to further improve retrieval performance [38, 20, 34, 77]. Some methods [77, 27, 34] leverage embeddings extracted by MLLMs for similarity-based retrieval, while LamRA

[38] employs MLLMs to rerank candidates and identify the best match. In this work, we propose the first R1-style MLLM for multimodal retrieval, leveraging the strong reasoning capabilities acquired through reinforcement learning to significantly enhance retrieval effectiveness.

**MLLM.** MLLMs [37, 29, 57, 79] extend the capabilities of LLMs to the multimodal domain and have achieved remarkable success across various tasks, such as visual question answering [25, 52, 46] and image segmentation [26, 49]. To improve their efficiency, recent studies [74, 70, 65, 32, 54] have explored reducing the number of input tokens to lower computational costs. For example, [74] introduces a compression module that reduces visual tokens while performing early fusion of image and instruction information. [70] compresses video tokens to support long-context video inputs. Our work also incorporates a token compression mechanism to accommodate more candidate inputs, but it features the following key novelties compared to prior methods: (1) Our compression mechanism is the first specifically tailored for universal retrieval, including a novel network architecture and training strategy to ensure that compressed tokens preserve retrieval-relevant information. (2) We introduce a novel details inspection mechanism that enables the MLLM to automatically identify and incorporate the full, uncompressed token sequences of challenging candidates during the CoT process, thereby effectively mitigating the negative impact of information loss caused by token compression.

**LLM and MLLM Reasoning.** Enhancing the reasoning capabilities of LLMs and MLLMs is a key direction for AI research. Early work [64, 60, 75, 58, 85, 19] typically employed fixed-format Chain-of-Thought (CoT) prompting to enable LLM reasoning. Inspired by the success of DeepSeek-R1 [14], many recent methods have adopted reinforcement learning to optimize the reasoning abilities of LLMs [61, 67] and MLLMs [16, 56, 71, 66, 53, 83, 17]. In this work, we propose the first R1-style reasoning MLLM framework for multimodal retrieval, introducing several task-specific innovations, including a reasoning activation training strategy using a synthesized CoT retrieval dataset, and a novel reward function that incorporates a curriculum-based token efficiency constraint. These designs significantly improve both the effectiveness and efficiency of MLLM-based multimodal retrieval.

## 3 Method

### 3.1 Preliminaries and Overview

This paper proposes a novel framework, Retrv-R1, to tackle the task of universal retrieval. In this task, given a query $q$ of any modality (text, image, or interleaved formats), the goal is to retrieve the most relevant sample from a set $\Omega = \{c_n\}_{n=1}^{N}$ with $N$ candidates. Following [38], Retrv-R1 adopts a two-stage approach to improve retrieval efficiency and effectiveness through a coarse-to-fine process. In the first stage, embeddings are generated for $q$ and each candidate $c_n$ using an MLLM $\phi$. The top-K candidates with the highest embedding similarity to $q$ are then selected, forming a subset denoted as $C = \{c_k\}_{k=1}^{K}$. In the second stage, another MLLM $\theta$ is employed to further refine the retrieval by identifying the best-matching item from $C$, yielding the final retrieval result $\hat{c} = \theta(q, C)$. We adopt the same method as [38] for constructing the embedding model $\phi$ in the first stage. In the following sections, we primarily illustrate the proposed new method for building, training, and applying the second-stage selection model $\theta$ within our Retrv-R1 framework.

### 3.2 Baseline Method and Shortcomings

The recent success of reasoning LLMs and MLLMs motivates us to explore the potential of chain-of-thought (CoT) reasoning for enhancing MLLM-based retrieval. As an initial attempt, we first construct a baseline method by directly fine-tuning Qwen2.5-VL on the retrieval dataset using Group Relative Policy Optimization (GRPO) [14]. In this approach, the model receives the following prompt as input: `Please select the candidate that best matches <query> and tell me its index. Candidate 1: <`$c_1$`>, Candidate 2: <`$c_2$`>, ..., Candidate K: <`$c_K$`>. Please think it step-by-step.` A set of generated CoTs $\{o_1, o_2, \ldots, o_G\}$ is randomly sampled, and the model is optimized by maximizing the following objective:

$$
\begin{aligned}
\mathcal{J}_{\text{GRPO}}(\theta) = \mathbb{E}_{q \sim P(Q), \{o_i\}_{i=1}^{G} \sim \pi_{\theta_{\text{old}}}(O|q)} & \left[ \frac{1}{G} \sum_{i=1}^{G} \min \left( \frac{\pi_\theta(o_i)}{\pi_{\theta_{\text{old}}}(o_i)} A_i, \right. \right. \\
& \left. \left. \text{clip} \left( \frac{\pi_\theta(o_i)}{\pi_{\theta_{\text{old}}}(o_i)}, 1 - \epsilon, 1 + \epsilon \right) A_i \right) - \beta D_{\text{KL}} \left( \pi_\theta \| \pi_{\text{ref}} \right) \right],
\end{aligned}
\tag{1}
$$

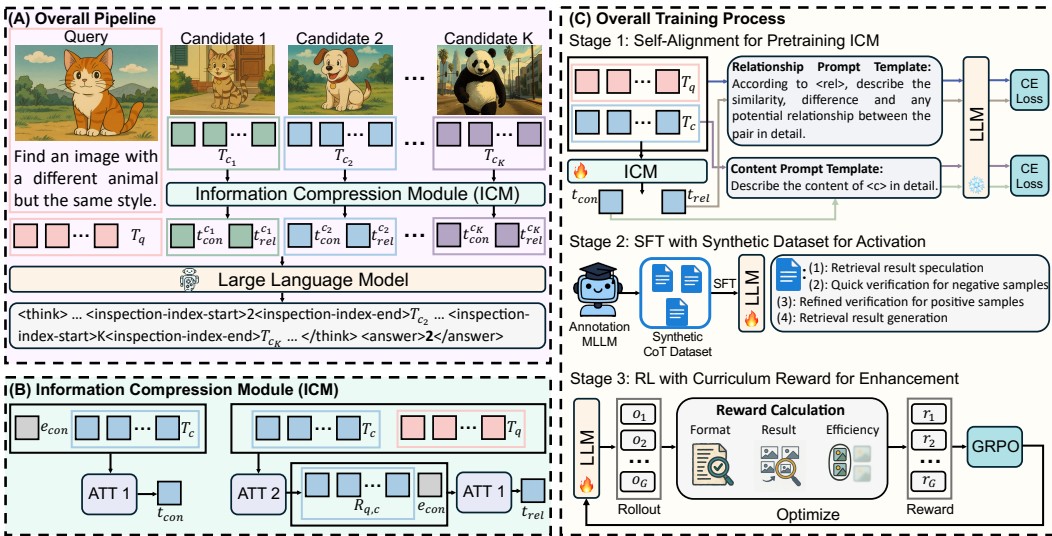

Figure 1: Illustration of (A) overall pipeline; (B) structure of information compression module (ICM); and (C) overall process of model training with three stages.

where $D_{\mathrm{KL}}$ refers to the KL divergence. $A_i = \frac{r_i - \mathrm{mean}(\{r_1, r_2, ..., r_G\})}{\mathrm{std}(\{r_1, r_2, ..., r_G\})}$ is the average of the group rewards $\{r_i\}_{i=1}^G$, where $r_i = 1$ if the final retrieval result $\hat{c}_i$ of $o_i$ is correct otherwise $r_i = 0$. All other hyperparameters in Eq.1 follow the same settings as [16]. As discussed in the introduction, this naive RL-trained model exhibits significant shortcomings in effectiveness, due to the convergence difficulties and training instability associated with direct RL-based optimization, as well as high computational costs caused by the large number of tokens from multiple candidates and fine-grained CoT reasoning. To address these challenges, Retrv-R1 extends the *model architecture* of the standard MLLM to improve retrieval efficiency, and introduces a new *training strategy* to enhance retrieval effectiveness. In the following Sec.3.3 and 3.4, we introduce these two key innovations, respectively.

### 3.3 Information Compression Module

We first design a simple but effective Information Compression Module (ICM) based on an intuitive idea: reducing the number of each $c_k$'s tokens for the language model (LM)'s inputs, thereby leaving sufficient context space for the complex and token-intensive CoT reasoning. As shown in Fig.1(A), the ICM is placed before the LM of the MLLM. Inspired by [74], given the original token sequence $T_{c_k}$ for each candidate $c_l$, ICM uses a learnable embedding $e_{con}$ and a two-layer attention module $\mathrm{ATT}_1$ to compress the information into a single content token $t_{con}^{c_k}$. This process is expressed as:

$$t_{con}^{c_k} = \mathrm{ATT}_1\left(\mathbf{Q}_{e_{con}}, \mathbf{K}_{T_{c_k}}, \mathbf{V}_{T_{c_k}}\right), \tag{2}$$

where $\mathbf{Q}$, $\mathbf{K}$ and $\mathbf{V}$ refer to the query, key and value in the attention mechanism, respectively. In retrieval tasks, beyond the standalone content of each candidate $c_k$ captured by $t_{con}^{c_k}$, its correlation with the query $q$ (e.g., their differences and similarities) is also critical for the accurate matching. To capture this crucial information, we further generate a relationship token $t_{rel}^{c_k}$ for each $c_k$ as follows:

$$t_{rel}^{c_k} = \mathrm{ATT}_1\left(\mathbf{Q}_{e_{con}}, \mathbf{K}_{R_{q,c_k}}, \mathbf{V}_{R_{q,c_k}}\right), \text{ where } R_{q,c_k} = \mathrm{ATT}_2\left(\mathbf{Q}_{T_{c_k}}, \mathbf{K}_{T_q}, \mathbf{V}_{T_q},\right), \tag{3}$$

where $T_q$ denotes the token sequence of query $q$ and $\mathrm{ATT}_2$ represents another 2-layer attention, $\mathbf{V}_{R_{q,c_k}}$ denotes the value features derived from $R_{q,c_k}$, where $R_{q,c_k}$ is the relationship feature between the query $q$ and the $k$-th candidate $c_k$. In this way, ICM compresses each $c_i$ into only two tokens, each carrying different types of information, thereby preserving key retrieval-relevant features while significantly reducing context usage.

**Self-Alignment for Pretraining ICM.** Despite its simple structure and intuitive functionality, training the ICM is non-trivial. Directly fine-tuning the ICM together with the LM in the MLLM often leads to convergence difficulties and suboptimal results. To address this challenge, we adopt a pretraining

mechanism inspired by BLIP-2 [29], aligning the output space of the ICM with the input space of the LM before jointly training them for the retrieval task. To ensure that the compressed tokens produced by the ICM effectively captures key retrieval-related information, we propose a self-alignment mechanism for pretraining with the following optimization function:

$$\mathcal{L}_{sa} = \mathbb{E}_{c_k} \left[ L_{ce} \left( \text{LM} \left( I_{con} \left[ t_{con}^{c_k} \right] \right), \text{LM} \left( I_{con} \left[ T_{c_k} \right] \right) \right) + L_{ce} \left( \text{LM} \left( I_{rel} \left[ t_{rel}^{c_k} \right] \right), \text{LM} \left( I_{rel} \left[ T_{c_k}; T_q \right] \right) \right) \right], \tag{4}$$

where $L_{ce}$ denotes the cross-entropy loss. $I_{con}$ and $I_{rel}$ are 2 instruction templates that prompt the LM to generate content and relationship descriptions, respectively. There details are as follows:

- $I_{con}$[`<`$c_k$`>`]: `Describe the content of <`$c_k$`> in detail.`
- $I_{rel}$[`<rel>`]: `According to <rel>, describe the similarity, difference and any potential relationship between the pair in detail.`

We optimize ICM using Eq.4 while keeping the LM frozen. This alignment constrains the tokens $t_{con}^{c_k}$ and $t_{rel}^{c_k}$ generated by the ICM to capture comprehensive information critical for the retrieval task, providing a preliminary pretraining of the ICM's functionality and thereby facilitating subsequent fine-tuning on the retrieval tasks. The overall training process will be detailed in Sec.3.5.

**Details Inspection Mechanism.** Through empirical experiments, we find that the condensed tokens $t_{con}^{c_i}$ and $t_{rel}^{c_i}$ are sufficient for the MLLM to determine the matching outcome between most candidates and the query. However, for some challenging candidates, these tokens are insufficient, as information loss caused by token compression can impair retrieval accuracy. To address this issue, we further propose a details inspection mechanism, where the MLLM autonomously identifies challenging candidates requiring "careful examination" during the CoT process and retrieves their full token sequence $T_{c_i}$ as supplementary information. This mechanism is simple and effective, requiring only the addition of two special tokens, `<inspection-index-start>` and `<inspection-index-end>`, to the MLLM's vocabulary. During the CoT process, the generation of `<inspection-index-start>` triggers the inspection operation. An index $idx$ is generated between the `<inspection-index-start>` and `<inspection-index-end>` tokens, after which the uncompressed token sequence $T_{c_{idx}}$ for $c_{idx}$ is appended following `<inspection-index-end>` as supplementary information. During training, we construct synthetic training samples to optimize the MLLM's ability to identify challenging candidates, and design a novel reward function to prevent excessive use of this mechanism, which could reduce efficiency. The details of these designs will be introduced in the following sections.

### 3.4 Model Training with Activation and Enhancement

After constructing the framework, the next key question is how to train the MLLM's CoT-based retrieval capability. To enhance effectiveness, inspired by [56, 16], we adopt a two-phase training strategy: SFT for reasoning initialization followed by RL for reasoning enhancement.

**Phase I: SFT with Synthetic Data.** We first perform SFT on CoT datasets to activate the model's basic reasoning abilities. To address the challenge that existing retrieval datasets do not provide CoT annotations, we sample 100K instances from M-BEIR and use Qwen2.5-VL-72B to synthesize a CoT dataset structured into four reasoning stages: *(1) Retrieval result speculation:* Based on the query, the model speculates the ideal retrieval result and generates its description. *(2) Quick verification for negative samples:* We find that most candidates can be easily identified as negative samples without requiring complex reasoning. To improve efficiency, we include the following text into the CoT label for the quick negative identification: "$\mathcal{N}$ `are clearly the negative samples`," Here, $\mathcal{N}$ refers to the set of clearly negative candidates identified by Qwen2.5-VL-72B. *(3) Refined verification for positive samples:* For the remaining candidates, a fine-grained reasoning process—including analyzing the candidate-query relationship and comparing different candidates—is conducted to identify the correct positive sample. At this stage, we further instruct Qwen2.5-VL-72B to select candidates that are difficult to judge and add the following text to the CoT: "`To facilitate better judgment, the full token sequences are provided for the following candidates:` `<inspection-index-start>idx1<inspection-index-end><`$T_{c_{idx1}}$`>, ...,` `<inspection-index-start>idxI<inspection-index-end><`$T_{c_{idxI}}$`>`". This trains the model to identify challenging candidates and perform the details inspection mechanism described in the previous section. *(4) Retrieval result generation:* Based on the above CoT reasoning, the final retrieval result is generated. An example of the synthesized CoT annotation is shown in Appendix Fig.3. More details on using Qwen2.5 for this data generation are provided in Appendix Sec.A.1.

**Phase II: RL with Curriculum Efficiency Constraint.** Through SFT on the synthesized dataset described above, the model acquires preliminary CoT reasoning capabilities for retrieval. We then fine-tune the model using RL to further enhance its reasoning performance and improve efficiency. Specifically, GRPO (Eq.1) is used as the optimization objective, and the reward model in this RL framework consists of two components: the formatting reward $r_f$ and the result-efficiency reward $r_r$. The formatting reward $r_f$ equals 1 if the CoT follows the correct structure "`<think>...</think><answer>...</answer>`" and the details inspection operation is formatted as "`<inspection-index-start>...<inspection-index-end>`". Otherwise, $r_f = 0$. The result-efficiency reward $r_r$ is computed as follows:

$$r_r = \mathbb{1}\left(\hat{c} = \hat{c}_{gt}\right)\left(1 - \lambda\frac{N_{ins}}{K}\right), \tag{5}$$

where $\hat{c}$ and $\hat{c}_{gt}$ denote the predicted and ground-truth retrieval results, respectively. $N_{ins}$ represents the number of `<inspection-index-start><inspection-index-end>` pairs in the generated CoT, i.e., the number of candidates for which the full token sequence is used. $K$ is the total number of candidates, and $\lambda$ is a weighting coefficient. This reward captures both the accuracy of the retrieval result and the frequency of using the token-consuming full sequences, thereby promoting improvements in both model effectiveness and retrieval efficiency.

We empirically find that directly setting a large value for $\lambda$ leads to suboptimal results. This may be because the MLLM is relatively weak in the early stages of training and thus relies more heavily on access to the full, information-rich token sequences $T_{c_k}$ to make accurate decisions. As a result, imposing a strong token efficiency constraint too early can hinder performance optimization. In contrast, as the MLLM becomes more effective in later stages, increasing the constraint to emphasize efficiency becomes more appropriate. Motivated by this observation, we propose a simple yet effective curriculum learning strategy, in which $\lambda$ is gradually increased throughout training. Specifically, we define $\lambda_i = i/N_{iter}$, where $\lambda_i$ denotes the value for $\lambda$ in the $i$-th training iteration and $N_{iter}$ is the total number of iterations. This creates a progressively stronger efficiency constraint schedule that improves model performance, as demonstrated by the experimental results in Table.7.

### 3.5 Overall Process of Training and Inference

After introducing the proposed new methods, we then present the overall process for model training and inference in this section. As shown in Fig.1(c), the training pipeline consists of three stages: First, the MLLM is frozen, and the Information Compression Module (ICM) is pretrained on M-BEIR using the methods described in Sec.3.3 and Eq.4. Next, we perform SFT to jointly train the ICM and MLLM on the synthesized dataset using cross-entropy loss. Finally, both the ICM and MLLM are fine-tuned via the reinforcement learning framework introduced in Sec.3.4, using 10K challenging samples from M-BEIR on which the SFT model fails to produce accurate results.

During inference, we treat the combination of the MLLM and ICM as the second-stage model $\theta$ described in Sec.3.1. It takes as input the top-K candidates preselected by the first-stage model $\phi$ (see Sec.3.1 for details) and produces the final retrieval result.

## 4 Experiments

### 4.1 Implementation Details

All experiments in our work are conducted on 16 A100 GPUs, using Qwen2.5-VL-3B and Qwen2.5-VL-7B pretrained through the methods in LamRA [38] as the MLLM $\theta$. As described in Sec.3.5, the overall training process consists of three stages. Both the first stage (pretraining) and the second stage (SFT) are trained for 1 epoch with a batch size of 64 and a learning rate of 1e-5. The third stage (RL) is trained for 1 epoch with a learning rate of 1e-6, using 8 rollouts and $\beta = 0.2$ in Eq.1. During training, the vision encoder remains frozen in all stages, the ICM is fully updated, and the language model is fine-tuned using LoRA. The number of candidates $K$ input to the MLLM $\theta$ is set to 50.

### 4.2 Main Results

**Comparison of Effectiveness.** We first evaluate the effectiveness of Retrv-R1 on the M-BEIR test set. The results on the Recall@K metric are presented in Table.1, covering 16 sub-tasks across 8

Table 1: **Comparison with other methods on M-BEIR test set.** R@K refers to the Recall@K metric. $q^t$, $q^i$, $c^t$ and $c^i$ denote the text query, image query, text candidates and image candidates, respectively.

| Methods | $q^t \rightarrow c^i$ | | | $q^t \rightarrow c^t$ | $q^t \rightarrow (c^i,c^t)$ | | $q^i \rightarrow c^t$ | | | $q^i \rightarrow c^i$ | $(q^i,q^t) \rightarrow c^t$ | | $(q^i,q^t) \rightarrow c^i$ | | $(q^i,q^t) \rightarrow (c^i,c^t)$ | | Avg |
|---|---|---|---|---|---|---|---|---|---|---|---|---|---|---|---|---|---|
| | VN | COCO | F200K | WebQA | EDIS | WebQA | VN | COCO | F200K | NIGHTS | OVEN | InfoS | FIQ | CIRR | OVEN | InfoS | |
| | R@5 | R@5 | R@10 | R@5 | R@5 | R@5 | R@5 | R@5 | R@10 | R@5 | R@5 | R@5 | R@10 | R@5 | R@5 | R@5 | |
| CLIP-L [48] | 43.3 | 61.1 | 6.6 | 36.2 | 43.3 | 45.1 | 41.3 | 79.0 | 7.7 | 26.1 | 24.2 | 20.5 | 7.0 | 13.2 | 38.8 | 26.4 | 32.5 |
| SigLIP [69] | 30.1 | 75.7 | 36.5 | 39.8 | 27.0 | 43.5 | 30.8 | 88.2 | 34.2 | 28.9 | 29.7 | 25.1 | 14.4 | 22.7 | 41.7 | 27.4 | 37.2 |
| BLIP [30] | 16.4 | 74.4 | 15.9 | 44.9 | 26.8 | 20.3 | 17.2 | 83.2 | 19.9 | 27.4 | 16.1 | 10.2 | 2.3 | 10.6 | 27.4 | 16.6 | 26.8 |
| BLIP2 [29] | 16.7 | 63.8 | 14.0 | 38.6 | 26.9 | 24.5 | 15.0 | 80.0 | 14.2 | 25.4 | 12.2 | 5.5 | 4.4 | 11.8 | 27.3 | 15.8 | 24.8 |
| UniIR-BLIP$_{FF}$ [59] | 23.4 | 79.7 | 26.1 | 80.0 | 50.9 | 79.8 | 22.8 | 89.9 | 28.9 | 33.0 | 41.0 | 22.4 | 29.2 | 52.2 | 55.8 | 33.0 | 46.8 |
| UniIR-CLIP$_{SF}$ [59] | 42.6 | 81.1 | 18.0 | 84.7 | 59.4 | 78.7 | 43.1 | 92.3 | 18.3 | 32.0 | 45.5 | 27.9 | 24.4 | 44.6 | 67.6 | 48.9 | 50.6 |
| Qwen2.5-VL-3B [2] | 36.0 | 67.8 | 16.1 | 69.5 | 45.2 | 61.7 | 23.3 | 82.3 | 12.0 | 20.9 | 36.7 | 22.3 | 24.3 | 53.5 | 56.4 | 49.8 | 42.4 |
| Qwen2.5-VL-7B [2] | 40.2 | 71.9 | 20.3 | 71.9 | 49.4 | 64.5 | 29.3 | 84.6 | 19.4 | 25.5 | 42.4 | 32.1 | 25.0 | 55.1 | 60.8 | 54.9 | 46.7 |
| Vision-R1-7B [16] | 41.9 | 75.0 | 22.0 | 70.6 | 51.3 | 69.1 | 35.4 | 85.1 | 22.4 | 25.9 | 48.8 | 44.0 | 29.2 | 57.7 | 66.2 | 59.0 | 50.2 |
| VLM-R1-7B [53] | 40.5 | 77.2 | 22.5 | 72.3 | 50.0 | 67.9 | 36.2 | 86.3 | 20.9 | 26.4 | 48.8 | 37.5 | 29.9 | 57.4 | 64.0 | 62.3 | 50.0 |
| MM-Embed-7B [34] | 41.0 | 71.3 | 17.1 | 95.9 | 68.8 | 85.0 | 41.3 | 90.1 | 18.4 | 32.4 | 42.1 | 42.3 | 25.7 | 50.0 | 64.1 | 57.7 | 52.7 |
| LamRA-7B [38] | 48.0 | 85.2 | 32.9 | 96.7 | 75.8 | 87.7 | 48.6 | 92.3 | 36.1 | 33.5 | 59.2 | 64.1 | 37.8 | 63.3 | 79.2 | 78.3 | 63.7 |
| Retrv-R1-3B | 46.6 | 83.0 | 34.4 | 96.7 | 77.9 | 88.2 | 48.0 | 92.7 | 34.5 | 34.7 | 64.3 | 70.0 | 45.2 | 67.9 | 83.3 | 80.0 | 65.5 |
| Retrv-R1-7B | **50.5** | **86.7** | **39.3** | **97.4** | **82.1** | **89.5** | **51.4** | **94.1** | **39.8** | **39.5** | **69.0** | **75.6** | **49.5** | **72.3** | **86.6** | **83.7** | **69.2** |

Table 2: **Comparison of effectiveness and efficiency. ITR**: The ratio of the inference time required by a method compared to the time required by Retrv-R1-7B ($K = 50$). **GMR**: The ratio of the GPU memory usage required by a method compared to that of Retrv-R1-7B ($K = 50$).

| Methods | $(q^i,q^t) \rightarrow c^i$ (CIRR) | | | | | | | | | | | | | | |
|---|---|---|---|---|---|---|---|---|---|---|---|---|---|---|---|
| | $K = 10$ | | | $K = 20$ | | | $K = 30$ | | | $K = 40$ | | | $K = 50$ | | |
| | R@5 ↑ | ITR ↓ | GMR ↓ | R@5 ↑ | ITR ↓ | GMR ↓ | R@5 ↑ | ITR ↓ | GMR ↓ | R@5 ↑ | ITR ↓ | GMR ↓ | R@5 ↑ | ITR ↓ | GMR ↓ |
| Qwen2.5-VL-7B [2] | 50.5 | 0.53 | 0.64 | 52.6 | 1.10 | 1.02 | 53.1 | 2.52 | 1.58 | 54.4 | 3.63 | 1.97 | 55.1 | 4.79 | 2.44 |
| Vision-R1-7B [16] | 51.2 | 1.81 | 1.30 | 53.7 | 3.14 | 1.75 | 54.8 | 4.58 | 2.25 | 57.0 | 5.93 | 2.87 | 57.7 | 7.23 | 3.28 |
| VLM-R1-7B [53] | 51.3 | 1.96 | 1.39 | 53.1 | 3.28 | 1.80 | 54.7 | 4.63 | 2.27 | 56.4 | 6.10 | 2.94 | 57.4 | 7.70 | 3.45 |
| LamRA-Rank-L-7B [38] | 62.7 | 0.56 | 0.65 | 64.7 | 1.10 | 1.03 | 65.5 | 2.57 | 1.60 | 65.9 | 3.70 | 1.97 | 66.2 | 4.98 | 2.46 |
| Retrv-R1-7B | **67.9** | **0.17** | **0.29** | **69.9** | **0.26** | **0.35** | **70.8** | **0.46** | **0.69** | **71.8** | **0.76** | **0.85** | **72.3** | **1.00** | **1.00** |

different combinations of query and candidate formats. To assess the scalability and generalization of our approach, we report results for both Retrv-R1-3B and Retrv-R1-7B. The compared methods include: (1) zero-shot general-purpose models, such as BLIP-2 [29] and Qwen-VL [57]; (2) prior R1-style reasoning MLLMs, including Vision-R1 [16] and VLM-R1 [53]; and (3) MLLMs specifically designed and trained for retrieval tasks, including MM-Embed [34] and LamRA [38]. As shown in Table.1, Retrv-R1 consistently achieves state-of-the-art (SOTA) performance across all scenarios. It is important to note that: (1) Even with only 3B parameters, Retrv-R1-3B still outperforms larger models like MM-Embed-7B and LamRA-7B in most cases. (2) On challenging tasks such as $(q^i,q^t) \rightarrow c^i$ on FIQ, Retrv-R1 maintains strong performance and outperforms prior advanced methods by an even larger margin. These results demonstrate the strong retrieval capabilities of Retrv-R1 across diverse data formats, highlighting its high effectiveness and universality.

**Comparison of Efficiency.** Table.2 presents efficiency comparisons by using the metrics of ITR and GMR, which respectively calculate the ratio of average inference time and GPU memory usage of a method compared to our Retrv-R1-7B with $K = 50$. The evaluation is conducted on the $(q^i,q^t) \rightarrow c^i$ setting from the M-BEIR test set. Two types of methods are included for comparison: (1) standard MLLMs and other R1-style MLLMs not designed for retrieval, including Qwen2.5-VL [2], Vision-R1 [16] and VLM-R1 [53]; and (2) LamRA [38] with listwise reranking (LamRA-Rank-L), which operates similarly to Retrv-R1 by retrieving from the top-K preselected candidates. As shown in Table.2, increasing the number of input candidates ($K$) generally improves retrieval performance, but it also leads to higher token consumption and computational cost. Moreover, a too large $K$ may exceed the memory limits of a single GPU card, which restricts other methods to smaller $K$ values in practical applications. In contrast, benefiting from the proposed ICM and the efficiency constraint in the RL reward, Retrv-R1 achieves significantly lower consumption of both inference time and GPU memory. Also note that: (1) Under the same $K$, Retrv-R1 achieves the best performance with the lowest computational cost. (2) Even with more input candidates ($K = 50$), Retrv-R1's inference time still remains lower than that of Qwen2.5-VL and LamRA-Rank-L with only $K = 20$. These results demonstrate the substantial advantages of Retrv-R1 in both effectiveness and efficiency.

Table 3: **Experimental results on unseen datasets.** $q^{\text{dialog}}$ and $(q^i \oplus q^t)$ refer to the dialog queries and multi-interleaved image-text queries, respectively.

| Methods | $(q^i, q^t) \to c^i$ | | $q^{\text{dialog}} \to c^i$ | $(q^i \oplus q^t) \to c^i$ | |
| --- | --- | --- | --- | --- | --- |
| | CIRCO | GeneCIS | VisD | VIST | MT-FIQ |
| | MAP@5 | R@1 | R@1 | R@1 | R@5 |
| CLIP-L [48] | 4.0 | 13.3 | 23.7 | 0.6 | 17.7 |
| UniIR-CLIP [59] | 12.5 | 16.8 | 26.8 | 0.6 | 39.4 |
| E5-V [20] | 24.8 | 18.5 | 54.6 | 10.0 | 19.2 |
| MagicLens-L [72] | 29.6 | 16.3 | 28.0 | 3.3 | 22.6 |
| MM-Embed-7B [34] | 35.5 | 22.9 | 64.7 | 25.7 | 59.0 |
| LmaRA-7B [38] | 42.8 | 24.8 | 70.9 | 28.6 | 63.9 |
| **Retrv-R1-7B** | **47.8** | **29.2** | **74.2** | **30.9** | **69.6** |

Table 4: **Experimental results on held-out tasks.** $*$ indicates that training is performed on the remaining tasks, w/o any exposure to the three held-out tasks.

| Methods | $q^i \to c^i$ | $(q^i, q^t) \to c^t$ | | $(q^i, q^t) \to (c^i, c^t)$ | | Avg. |
| --- | --- | --- | --- | --- | --- | --- |
| | NIGHTS R@5 | OVEN R@5 | InfoS R@5 | OVEN R@5 | InfoS R@5 | |
| *Supervised* | | | | | | |
| UniIR-BLIP$_{FF}$ [59] | 33.0 | 41.0 | 22.4 | 55.8 | 33.0 | 37.0 |
| UniIR-CLIP$_{SF}$ [59] | 32.0 | 45.5 | 27.9 | 67.6 | 48.9 | 44.4 |
| *Zero-shot* | | | | | | |
| Qwen2.5-VL-7B [2] | 20.3 | 38.5 | 40.4 | 53.6 | 44.9 | 39.5 |
| Vision-R1-7B [16] | 22.9 | 39.8 | 42.9 | 57.4 | 46.5 | 41.9 |
| LamRA-7B* [38] | 29.2 | 46.9 | 54.2 | 65.1 | 59.1 | 50.9 |
| **Retrv-R1-7B*** | **35.5** | **57.3** | **65.0** | **74.6** | **71.7** | **60.8** |

Table 5: **Experimental results on multimodal sequential recommendation task.** Three subsets from the Amazon Review benchmark are evaluated. HR@K and N@K refer to the hit ratio and normalized discounted cumulative gain. # denotes fine-tuned model.

| Methods | Sports | | Beauty | | Toys | |
| --- | --- | --- | --- | --- | --- | --- |
| | HR@10 | N@10 | HR@10 | N@10 | HR@10 | N@10 |
| S3-Rec [78] | 3.85 | 2.04 | 6.47 | 3.27 | 7.00 | 3.76 |
| E4SRec [33] | 4.10 | 2.37 | 7.58 | 4.35 | 7.98 | 4.79 |
| P5 [11] | 4.60 | 3.36 | 6.64 | 4.29 | 7.09 | 5.87 |
| VIP5 [12] | 4.75 | 3.65 | 6.77 | 4.67 | 7.49 | 6.04 |
| POD [31] | 5.76 | 4.19 | 6.88 | 4.43 | 7.42 | 6.10 |
| ICSRec [47] | 5.65 | 3.35 | 9.60 | 5.79 | 10.55 | 6.57 |
| Retrv-R1-3B | 7.13 | 4.72 | 8.90 | 5.65 | 9.69 | 6.11 |
| Retrv-R1-3B$^{\#}$ | **9.95** | **6.39** | **12.71** | **7.78** | **12.89** | **7.53** |

Table 6: **Experimental results of ablation study.** **ITR** denotes the ratio of the inference time required by a method compared to the time required by Retrv-R1-3B. **DIM** refers to the details inspection mechanism introduced in Sec.3.3.

| Methods | $(q^i, c^i) \to c^i$ CIRR | | $(q^i, q^t) \to (c^i, c^t)$ OVEN | |
| --- | --- | --- | --- | --- |
| | R@5↑ | ITR↓ | R@5↑ | ITR↓ |
| Retrv-R1-3B | 67.9 | 1.00 | 83.3 | 1.00 |
| Retrv-R1-3B w/o ICM | 68.8 | 7.40 | 84.4 | 7.99 |
| Retrv-R1-3B w/o $t_{con}$ | 60.7 | 0.97 | 77.6 | 0.95 |
| Retrv-R1-3B w/o $t_{rel}$ | 64.7 | 0.97 | 80.7 | 0.96 |
| Retrv-R1-3B w/o self-alignment | 64.3 | 1.05 | 80.7 | 1.02 |
| Retrv-R1-3B w/o DIM | 62.3 | 0.87 | 78.1 | 0.81 |
| Retrv-R1-3B w/o SFT stage | 63.0 | 1.60 | 79.4 | 1.85 |
| Retrv-R1-3B w/o RL stage | 61.1 | 1.39 | 78.4 | 1.75 |

## 4.3 Extended Experiments to Validate Generalization

We conduct the following three experiments to further evaluate the generalization ability of Retrv-R1: (1) Evaluate Retrv-R1 on **unseen retrieval datasets** that were not used during training. (2) Assess Retrv-R1 on **unseen retrieval tasks** by removing certain tasks from training set and evaluating the retrained model on these held-out tasks. (3) Test Retrv-R1's performance on **multimodal recommendation tasks** (see Appendix.A.2 for method details). The results are shown in Tables 3, 4, and 5, respectively. Retrv-R1 consistently achieves strong performance across all settings, demonstrating its robust generalization to diverse data types and tasks. Notably, as shown in Table.5, Retrv-R1 performs effectively on multimodal recommendation even without any fine-tuning, and becomes even stronger with SOTA results after task-specific fine-tuning, outperforming other task-specific models. These results highlight the great potential of Retrv-R1 as a general framework for broader domains.

## 4.4 Ablation Study

We further conduct an ablation study to evaluate the effectiveness of the following components and designs in our framework: (1) **ICM** for token compression to enhance efficiency; (2) **Two compressed tokens** $t_{con}$ and $t_{rel}$ extracted by the ICM; (3) **Self-alignment** for pretraining the ICM; (4) The **details inspection mechanism (DIM)** for handling challenging candidates; (5) **SFT** for activating the model's basic reasoning abilities; and (6) **RL** for reasoning enhancement. The results shown in Table.6 indicate that removing any of these components significantly degrades either performance or efficiency, validating their importance. Notably, while ICM may slightly affect performance due to feature compression, it substantially improves retrieval efficiency by reducing inference time. Similarly, the details inspection mechanism introduces only a minor computational overhead while significantly boosting accuracy. Additionally, we examine (7) replacing the **curriculum strategy** with a fixed $\lambda$ in Eq.5. As shown in Table.7, this modification leads to a notable drop in retrieval accuracy, further demonstrating the effectiveness of our design in reward settings. Note that we report results on the $(q^i, q^t) \to c^i$ task on CIRR and the $(q^i, q^t) \to (c^i, c^t)$ task on OVEN are reported, as these are highly challenging settings that can effectively evaluate the performance of different methods under difficult scenarios. Appendix Table 10 presents the

Table 7: **Experimental results of ablation study for** $\lambda$ **in Eq.5. ITR** denotes the ratio of the inference time required by a method compared to the time required when using the curriculum efficiency constraint.

| Methods | $(q^i, c^i) \to c^i$ CIRR | | $(q^i, q^t) \to (c^i, c^t)$ OVEN | |
|---|---|---|---|---|
| | R@5↑ | ITR↓ | R@5↑ | ITR↓ |
| Curriculum Efficiency Constraint | 67.9 | 1.00 | 83.3 | 1.00 |
| Fixed $\lambda = 0.5$ | 66.6 | 1.23 | 82.6 | 1.39 |
| Fixed $\lambda = 0.75$ | 65.2 | 1.01 | 81.6 | 0.96 |
| Fixed $\lambda = 1$ | 64.9 | 0.95 | 81.1 | 0.87 |
| Fixed $\lambda = 1.5$ | 64.1 | 0.87 | 80.0 | 0.80 |

Table 8: **Comparison of RAG capabilities on the Knowledge-based Visual Question Answering (KVQA) tasks. PR@K** and **ACC** denote the precision and accuracy metrics, respectively.

| Method | OKVQA [41] | Infoseek [6] | E-VQA [42] |
|---|---|---|---|
| *Retrieval (PR@5)* | | | |
| PreFLMR [36] | 70.9 | 62.1 | 73.7 |
| LamRA-7B [38] | 89.0 | 73.4 | 75.0 |
| Retrv-R1-7B | **91.7** | **77.8** | **79.8** |
| *VQA (ACC)* | | | |
| RA-VQAv2 w/ PreFLMR [35] | 61.9 | 30.7 | 54.5 |
| LamRA-7B [38] | 64.3 | 28.8 | 56.2 |
| Retrv-R1-7B | **66.0** | **31.5** | **58.4** |

averaged ablation results across all tasks on M-BEIR, where each component of Retrv-R1 continues to show significant contributions, further demonstrating the effectiveness and generality of our design choices in the framework.

## 4.5 Experiments on RAG

Retrieval-Augmented Generation (RAG) is an important application area of retrieval, which enhances the capabilities of LLMs by retrieving relevant external information as supplementary context. We evaluate the proposed Retrv-R1

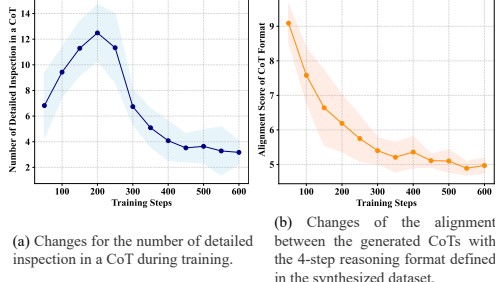

(a) Changes for the number of detailed inspection in a CoT during training.

(b) Changes of the alignment between the generated CoTs with the 4-step reasoning format defined in the synthesized dataset.

Figure 2: Analysis results of RL fine-tuning.

within RAG settings, with results presented in Table.8. Following [38], the experiments are conducted on three Knowledge-based Visual Question Answering (KVQA) tasks, where the MLLM is jointly trained on both the retrieval and KVQA. As shown in Table.8, Retrv-R1 achieves the best performance in both retrieval accuracy and VQA accuracy, demonstrating its strong effectiveness when applied in the RAG framework, highlighting the broad applicability of our method.

## 4.6 Analysis of RL Fine-tuning

To gain deeper insights from our framework, we further analyze the RL fine-tuning stage in our method and observe the following patterns:

**(1) Effectiveness before efficiency:** In the synthetic data used for SFT, we employ a details inspection mechanism to identify and use the full features for challenging candidates. During RL stage, as shown in Fig.2(a), we observe the number of detailed inspections in each generated CoT initially increases while then declines throughout the training process. This could be because, in the early phase of RL fine-tuning, the model prioritizes optimizing accuracy through more fine-grained reasoning, resulting in more thorough inspections to enhance performance. Once a high level of accuracy is reached, the model shifts focus toward improving efficiency, adapting to the stronger efficiency constraint introduced in Eq.5. These observations highlight the effectiveness of our curriculum-based RL reward design for promoting efficiency.

**(2) Increasingly flexible reasoning processes:** We further evaluate the formats of the generated CoTs using Qwen2.5-VL-72B, scoring their alignment with the 4-step reasoning format defined in the synthesized dataset (on a scale from 0 to 10, with higher scores indicating better alignment). As shown in Fig.2(b), CoTs become progressively more flexible and diverse in structure as training progresses, no longer strictly following the fixed format of the synthetic dataset. In the later stages of RL fine-tuning, we observe the emergence of some new reasoning steps and strategies automatically learned by the model, such as: *(i) Self-reflection for reevaluation* — as shown in Appendix Fig.4, the MLLM occasionally performs self-reflection, identifying cases where a positive sample may have been mistakenly classified as negative during quick verification, and then re-examines it to reach the correct conclusion. *(ii) Indicating absence of correct results* — as shown in Appendix Fig.5, when none of the top-K candidates is correct, the model can explicitly signal this at the end of the CoT and suggest including more candidates for retrieval. These behaviors demonstrate the model's effective ability to learn flexible, powerful and robust reasoning capabilities through our RL framework.

# 5 Conclusion

This paper introduces Retrv-R1, the first R1-style MLLM framework designed for multimodal universal retrieval tasks. The framework incorporates several novel, task-tailored designs for both model architecture and training strategies, leading to substantial improvements in model effectiveness, inference efficiency and task generalization, as demonstrated by extensive experiments across multiple benchmarks. We regard Retrv-R1 as a key step toward leveraging the reasoning capabilities of MLLMs to enhance downstream multimodal tasks, laying a solid foundation for the development of general-purpose multimodal systems with advanced reasoning abilities.

## Acknowledgment

The research was partially supported by the RGC General Research Fund 11200323, NSFC/RGC JRS Project N_CityU198/24. We thank Mr. Liqun Liu and Mr. Peng Shu from Tencent for their collaborations, insightful discussions, and support with computational resources in this work.

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

# A  More Details of Method

## A.1  Details of Synthesizing Retrieval CoT Dataset

As discussed in Sec.3.4 of the main paper, before using RL for further optimization, we leverage Qwen2.5-VL-72B to synthesize a retrieval CoT dataset to activate the reasoning capabilities of the MLLM. Given a query-candidate set $\{q, c_1, c_2, ..., c_K\}$, the process of synthesizing a CoT involves the following steps:

**Step 1: Retrieval result speculation:** We provide Qwen2.5-VL-72B with the following prompt, instructing it to generate a speculative description of what an ideal retrieval result should look like. The generated description is denoted a $d_{rrs}$.

---

**Prompt for retrieval result speculation**

You are an assistant tasked with retrieving the candidate that best matches the input query. The query is: <$q$>. Please provide a detailed description of what the ideal retrieval result should look like.

---

**Step 2: Quick verification for negative samples:** We provide Qwen2.5-VL-72B with the following prompt, instructing it to identify which candidates can easily be classified as negative samples without the need for extensive step-to-step reasoning. The outputs of Qwen are the indexes of these candidates, denoted as $\{ns1, ns2, ..., nsN_{ns}\}$.

---

**Prompt for quick verification of negative samples**

You are an assistant tasked with retrieving the candidate that best matches the input query. The query is: <$q$>. Candidate 1: <$c_1$>, Candidate 2: <$c_2$>, ..., Candidate K: <$c_K$>. Some candidates are clearly not the correct retrieval results and can therefore be easily and directly identified as negative samples, without the need for complex step-by-step reasoning (e.g., carefully examining the candidate's details, using logical relationships for reasoning, or comparing different candidates to find the best option). Please identify these candidates, provide the reasons for classifying them as such, and output their indexes. Please ensure that these candidates can indeed be easily and directly classified as negative samples.

---

**Step 3: Identification for challenging candidates:** For candidates not classified as negative samples, we instruct Qwen2.5-VL-72B to further identify challenging candidates for which the compressed tokens are insufficient and the full features are therefore required. The generated indexes for these candidates are denoted as $\{cc1, cc2, ..., ccN_{cc}\}$. The prompt used for this step is:

---

**Prompt for identification of challenging candidates**

You are an assistant tasked with retrieving the candidate that best matches the input query. The query is: <$q$>. Candidate 1: <$c_{ns1}$>, Candidate 2: <$c_{ns2}$>, ..., Candidate K: <$c_{nsK}$>. Among these candidates, some are particularly challenging and require a detailed examination, as they cannot be accurately assessed based solely on their rough summary or compressed features. Please identify these challenging candidates, provide the reasons for classifying them as such, and output their indexes. If no such challenging candidates exist, you can return an empty set. Please do not output an excessive number of challenging candidates; ensure that providing detailed information for these candidates is truly necessary, while the remaining candidates can be processed using only their general, compressed features.

---

**Step 4: Refined verification for positive samples :** For candidates not classified as negative samples, we instruct Qwen2.5-VL-72B to generate a fine-grained reasoning process, such as analyzing the candidate-query relationship and comparing different candidates, to help identify the correct positive sample. The generated reasoning process is denoted as $r_{vps}$. The prompt used in this step is:

---
**Prompt for refined verification of positive samples**

You are an assistant tasked with retrieving the candidate that best matches the input query. The query is: $<q>$. Candidate 1: $<c_{ns1}>$, Candidate 2: $<c_{ns2}>$, ..., Candidate K: $<c_{nsK}>$. Please generate a step-by-step, fine-grained reasoning process to identify the correct retrieval result $<gt>$ from these candidates. The reasoning process can follow any effective flow structure, such as, but not limited to, describing key details of the candidates, analyzing the candidate-query relationship, and comparing different candidates. Please ensure that the generated reasoning is comprehensive, logically coherent, and leads to the correct result.

---

**Step 5: Integrating into a Complete CoT:** We combine the information obtained from the previous steps into a CoT with the following template:

---
**Template of the synthetic dataset**

To assist the user in identifying the correct retrieval result, I first need to infer the characteristics of an ideal result. Based on the user-provided query, an ideal retrieval result should be: $d_{rrs}$. Next, I will analyze and evaluate the candidates. First, $\{ns1, ns2, ..., nsN_{ns}\}$ are clearly the negative samples and not the correct retrieval result. For the remaining candidates, to facilitate better judgment, the full token sequences are provided for the following candidates: <inspection-index-start>$cc1$<inspection-index-end>$<T_{c_{cc1}}>$, ..., <inspection-index-start>$cc_{N_{cc}}$<inspection-index-end>$<T_{cc_{N_{cc}}}>$". Next, I will perform detailed reasoning on these candidates: $r_{vps}$. Based on the reasoning above, the correct retrieval result is $<gt>$.

---

### A.2 Details of Applying Retrv-R1 to Multimodal Recommendation

As discussed in Sec.4.3 and Table.5 of the main paper, we evaluate Retrv-R1 on an out-of-domain multimodal recommendation task, where its outstanding performance demonstrates the strong generalization ability of Retrv-R1. In this experiment, Retrv-R1 is applied to multimodal recommendation using some simple extensions but without any modifications to the model structure. Specifically, we use the ICM (Sec.3.3) to generate a content token from each item of the human behavior data. All these tokens are concatenated to form the query in our method. The LLM's input instruction is modified to: Please recommend an item to the user based on his past behavior. The user's past behavior is: <query>. The candidate items include: Candidate 1 $<c_1>$, ..., Candidate K $<c_K>$. Please output the index of the recommended item. The remaining components, such as candidate processing and result generation, follow the same strategy as the original method.

## B  More Experimental Results

**Experiments on Text-Only Retrieval.** Our Retrv-R1 is designed for universal retrieval, meaning it can handle not only multimodal retrieval tasks as discussed in the main paper, but also text-only retrieval tasks with a single modality. To demonstrate this, we conduct additional experiments on the BEIR dataset, a standard text-only retrieval suite. Following the PE-Rank protocol, we rerank the top-100 retrieved passages across eight BEIR subsets: COVID, NFCorpus, Touché, DBPedia, SciFact, Signal, News, and Robust. Table 9 presents a comparison of the average NDCG@10 scores across these subsets. As shown, Retrv-R1 achieves the best performance, further demonstrating its strong generalization ability and versatility across both multimodal and text-only retrieval tasks.

## C  Qualitative Examples

To illustrate our approach and demonstrate the effectiveness of the proposed Retrv-R1 more intuitively, we present two types of qualitative examples: (1) examples of the synthesized retrieval CoT data for SFT (Fig.3), and (2) examples of retrieval results along with the reasoning processes generated by Retrv-R1 (Fig.4, Fig.5, Fig.6 and Fig.7). These results showcase how Retrv-R1 achieves accurate retrieval through fine-grained reasoning, underscoring the high effectiveness of the proposed method.

Table 9: Comparison on BEIR.

| Methods | Average NDCG@10 |
|---|---|
| BM25 | 0.4380 |
| monoBERT | 0.4716 |
| monoT5 | 0.5136 |
| RankMistral | 0.4365 |
| PE-Rank | 0.4843 |
| **Retrv-R1** | **0.5267** |

Table 10: Average ablation results on all tasks in M-BEIR.

| Methods | Avg. Score on M-BEIR | ITR |
|---|---|---|
| Retrv-R1-3B | 65.5 | 1.00 |
| Retrv-R1-3B w/o ICM | 66.3 | 6.97 |
| Retrv-R1-3B w/o $t_{con}$ | 59.2 | 0.97 |
| Retrv-R1-3B w/o $t_{rel}$ | 63.2 | 0.97 |
| Retrv-R1-3B w/o self-alignment | 63.5 | 1.02 |
| Retrv-R1-3B w/o DIM | 60.9 | 0.89 |
| Retrv-R1-3B w/o SFT stage | 62.4 | 1.52 |
| Retrv-R1-3B w/o RL stage | 61.5 | 1.26 |

## D Discussion of Limitations

This work presents the first R1-style MLLM specifically designed for multimodal universal retrieval, achieving SOTA performance, high efficiency, and strong generalization ability. However, the proposed Retrv-R1 still has one limitation: the Information Compression Module (ICM) introduces a slight performance drop – -0.9% on $(q^i, q^t) \rightarrow c^i$ and -1.1% on $(q^i, q^t) \rightarrow (c^i, c^t)$ (see Table.6) – due to information loss caused by token compression. Nonetheless, given that ICM significantly reduces inference time by over $7\times$, we believe this minor trade-off in accuracy is entirely acceptable and well justified. Furthermore, our proposed details inspection mechanism helps alleviate this issue by identifying challenging samples and leveraging their full tokens when necessary. In future work, we plan to explore better paradigms to achieve both stronger performance and higher efficiency.

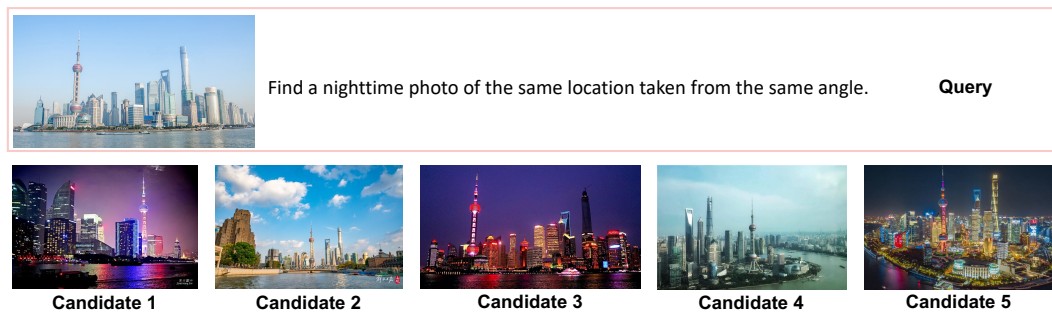

Figure 3: A qualitative example of the synthesized retrieval CoT data for SFT.

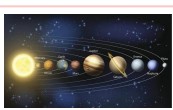 Find a text that can provide additional information for this image. **Query**

The Earth and the Moon form the Earth–Moon satellite system with a shared center of mass, or barycenter. This barycenter is 1,700 km (1,100 mi) (about a quarter of Earth's radius) beneath the Earth's surface. The Moon's orbit is slightly elliptical, with an orbital eccentricity of 0.055. The semi-major axis of the geocentric lunar orbit, called the lunar distance, is approximately 400,000 km (250,000 miles or 1.28 light-seconds), comparable to going around Earth 9.5 times. The Moon makes a complete orbit around Earth with respect to the fixed stars, its sidereal period, about once every 27.3 days. However, because the Earth–Moon system moves at the same time in its orbit around the Sun, it takes slightly longer, 29.5 days, to return to the same lunar phase, completing a full cycle, as seen from Earth. This synodic period or synodic month is commonly known as the lunar month and is equal to the length of the solar day on the Moon. Due to tidal locking, the Moon has a 1:1 spin–orbit resonance. This rotation–orbit ratio makes the Moon's orbital periods around Earth equal to its corresponding rotation periods. This is the reason for only one side of the Moon, its so-called near side, being visible from Earth. **Candidate 1**

The Milky Way or Milky Way Galaxy is the galaxy that includes the Solar System, with the name describing the galaxy's appearance from Earth: a hazy band of light seen in the night sky formed from stars in other arms of the galaxy, which are so far away that they cannot be individually distinguished by the naked eye. The Milky Way is a barred spiral galaxy with a D25 isophotal diameter estimated at $26.8 \pm 1.1$ kiloparsecs ($87,400 \pm 3,600$ light-years), but only about 1,000 light-years thick at the spiral arms (more at the bulge). Recent simulations suggest that a dark matter area, also containing some visible stars, may extend up to a diameter of almost 2 million light-years (613 kpc). The Milky Way has several satellite galaxies and is part of the Local Group of galaxies, forming part of the Virgo Supercluster which is itself a component of the Laniakea Supercluster. It is estimated to contain 100–400 billion stars and at least that number of planets. The Solar System is located at a radius of about 27,000 light-years (8.3 kpc) from the Galactic Center, on the inner edge of the Orion Arm, one of the spiral-shaped concentrations of gas and dust. The stars in the innermost 10,000 light-years form a bulge and one or more bars that radiate from the bulge. **Candidate 2**

A terrestrial planet, tellurian planet, telluric planet, or rocky planet, is a planet that is composed primarily of silicate, rocks or metals. Within the Solar System, the terrestrial planets accepted by the IAU are the inner planets closest to the Sun: Mercury, Venus, Earth and Mars. Among astronomers who use the geophysical definition of a planet, two or three planetary-mass satellites – Earth's Moon, Io, and sometimes Europa – may also be considered terrestrial planets. The large rocky asteroids Pallas and Vesta are sometimes included as well, albeit rarely. The terms "terrestrial planet" and "telluric planet" are derived from Latin words for Earth (Terra and Tellus), as these planets are, in terms of structure, Earth-like. Terrestrial planets are generally studied by geologists, astronomers, and geophysicists. Terrestrial planets have a solid planetary surface, making them substantially different from larger gaseous planets, which are composed mostly of some combination of hydrogen, helium, and water existing in various physical states. **Candidate 3**

The Nine-Planet Alignment, also known as nine-planet syzygy, is a rare astronomical event in which all nine celestial bodies traditionally considered planets in our solar system appear roughly aligned in the same region of the sky when viewed from Earth. This alignment does not mean the planets form a perfect straight line in space, as their orbits lie on different planes and distances, but rather that they cluster in the same general direction from Earth's perspective. Such an event is extremely rare, occurring only a few times over the course of several centuries, especially when including Pluto, whose highly tilted and elliptical orbit makes perfect alignment even more unlikely. While the alignment has no measurable gravitational or physical impact on Earth, it holds cultural and symbolic significance and has often been linked to myths, astrology, and pseudoscientific predictions. Some of the aligned planets may be visible to the naked eye during such an event, typically around dawn or dusk, while others like Uranus, Neptune, and Pluto require telescopes to observe. The last widely referenced nine-planet alignment occurred in May 2000, though not all planets were visible to the naked eye. **Candidate 4**

**Output from Retrv-R1:**
<think> The user's request is to find a text that can provide additional information about the given image. To do that, I first need to understand the content of the image. The image shows the structure of the solar system, including the eight major planets and Pluto, which was formerly considered the ninth planet. The Sun, a star at the center of the solar system, provides light and heat, and all planets revolve around it. The curved paths around each planet represent their orbits, illustrating how they travel around the Sun. The user is likely looking for a text that offers a more detailed introduction to the solar system, such as explanations of its formation or the characteristics of each planet. Let me now examine each candidate. I think Candidate 1, Candidate 2, and Candidate 4 are clearly not correct matches. <inspection-index-start>3<inspection-index-end>$T_{c_3}$. Candidate 3 mentions the solar system and includes Mercury, Venus, Earth, and Mars, which are all consistent with the image. However, the main focus of this text is on terrestrial planets, whereas the image also includes outer planets like Jupiter and Saturn. Therefore, Candidate 3 is not the right match either. It seems that none of the candidates fully meet the user's request—perhaps none of them are suitable? Let me re-examine both the query image and the candidates. Candidate 4 discusses the phenomenon of nine-planet alignment, which also occurs within the solar system and thus bears some relation to the image. In the query image, the nine celestial bodies are arranged in a straight line, which may represent a visual depiction of such an alignment. Therefore, Candidate 4 could potentially serve as relevant supplementary information for the query image. </think>
<answer>**4**</answer>

Figure 4: A qualitative example of the retrieval result generated from Retrv-R1.

Find an image suitable to be used as the illustration for this news:

In a moment that would forever be etched into football history, Argentina captain Diego Maradona hoisted the FIFA World Cup trophy high above his head at the Azteca Stadium, surrounded by ecstatic teammates and a sea of jubilant supporters. With their 3-2 victory over West Germany, Argentina secured their second World Cup title, capping off a tournament defined by brilliance, drama, and the unparalleled magic of their No. 10. The iconic image of Maradona, clad in the sky-blue and white stripes of Argentina, lifting the golden trophy under the blazing Mexican sun, became the defining symbol of the 1986 World Cup. Carried on the shoulders of his teammates, Maradona smiled triumphantly as photographers and fans rushed to capture the historic scene. Throughout the tournament, Maradona had been nothing short of sensational, delivering one of the most dominant individual performances in World Cup history. From his infamous "Hand of God" goal to the breathtaking solo run against England — later dubbed the "Goal of the Century" — Maradona led Argentina with unmatched flair and determination. In the final, Argentina took a 2-0 lead before West Germany mounted a comeback to level the score at 2-2. With just minutes remaining, Jorge Burruchaga scored the decisive goal after a sublime pass from Maradona, sealing the victory and sending the stadium into raptures. "This is for all of Argentina," Maradona declared after the match, emotionally overwhelmed. "We knew we could do it, and we did it with heart and soul." The image of Maradona lifting the World Cup remains a powerful emblem of footballing greatness — a snapshot of a genius at the peak of his powers, uniting a nation and inspiring generations to come.

**Query**

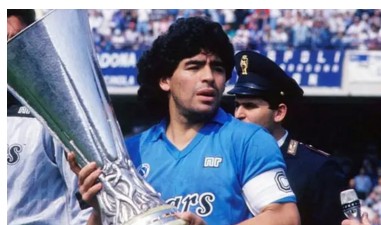

**Candidate 1**

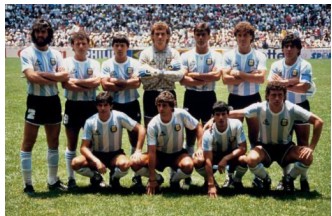

**Candidate 2**

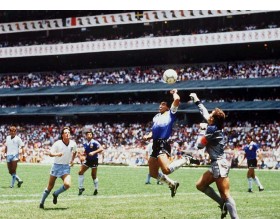

**Candidate 3**

**Output from Retrv-R1:**
<think> The user asks me to find an image suitable to be used as the illustration for the given news article. To begin, I need to summarize the main content of the article. The news focuses on Argentina's 3–2 victory over West Germany in the 1986 World Cup Final at Mexico's Azteca Stadium, where captain Diego Maradona delivered a legendary performance. His leadership throughout the tournament — highlighted by the "Hand of God" and the "Goal of the Century" — helped secure Argentina's second World Cup title. The article emphasizes Maradona's pivotal role in the team's triumph and his emotional display after the win, particularly the iconic moment when he lifted the trophy above his head. Therefore, the ideal illustration should clearly feature Maradona as the central figure, shown lifting the World Cup trophy at Azteca Stadium, with a strong emotional tone of victory and celebration. Images taken outside the stadium or those that fail to depict the trophy or the post-match celebration are not appropriate. Next, let me check all the candidate images. Candidate 1 shows Maradona holding a championship trophy and may seem relevant to the given news. Let me carefully check it: <inspection-index-start>1<inspection-index-end>$T_{c_1}$. However, upon closer inspection of this image, he is wearing a sky-blue jersey, which is the color of Napoli, not Argentina. The "N.R." logo on the chest confirms it is a kit from Ennerre, Napoli's kit sponsor at the time. The trophy in the image is tall and flared, with sculpted figures on the base — characteristics of the UEFA Cup rather than the world cup. Therefore, this image is from Napoli's UEFA Cup victory and does not match the content of the news article. Candidate 2 is a team photo of Argentina's starting lineup. While it is related to the 1986 World Cup, it does not prominently feature Maradona, nor does it capture the emotional post-victory moment described in the news. Candidate 3 shows the famous "Hand of God" moment. Although this is one of the most iconic scenes from the 1986 World Cup and is briefly mentioned in the article, it is not the main focus of the news. The image does not convey the celebration or culmination of the World Cup victory. In conclusion, while all three candidate images are related to Maradona, none of them captures the specific scene of him lifting the World Cup trophy in celebration, which is central to the given news. The most suitable illustration would be a photo of Maradona at the Azteca Stadium, wearing the Argentina jersey, lifting the FIFA World Cup trophy high above his head while being carried on the shoulders of his teammates, with a backdrop of a cheering crowd of tens of thousands. I recommend the user submit additional candidate images for further evaluation. </think>
<answer>**None**</answer>

Figure 5: A qualitative example of the retrieval result generated from Retrv-R1.

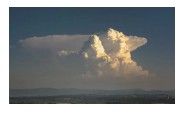 Find a text that explains the formation mechanism of this type of cloud. **Query**

Cumulonimbus clouds form through strong atmospheric convection. When warm, moist air near the Earth's surface rises due to solar heating or a lifting mechanism (such as a cold front or mountain slope), it cools and condenses to form clouds. If the atmosphere is unstable, the rising air continues to ascend, releasing latent heat that further fuels the updraft. As the cloud grows vertically, it can reach the top of the troposphere, where the updraft spreads out, forming an anvil-shaped top. This mature cumulonimbus cloud can produce severe weather such as thunderstorms, heavy rain, hail, strong winds, and even tornadoes. **Candidate 1**

Towering cumulonimbus clouds are typically accompanied by smaller cumulus clouds. The cumulonimbus base may extend several kilometres (miles) across or be as small as several tens of metres (yards) across and occupy low to upper altitudes within the troposphere - formed at altitude from approximately 200 to 4,000 m (700 to 10,000 ft). Normal peaks usually reach to as much as 12,000 m (39,000 ft), with unusually high ones typically topping out around 20,000 m (66,000 ft) and extreme instances claimed to be as high as 21,000 m (69,000 ft) or more. Well-developed cumulonimbus clouds are characterized by a flat, anvil shaped top (anvil dome), caused by wind shear or inversion at the equilibrium level near the tropopause. The shelf of the anvil may precede the main cloud's vertical component for many kilometres (miles) and be accompanied by lightning. Occasionally, rising air parcels surpass the equilibrium level (due to momentum) and form an overshooting top culminating at the maximum parcel level. When vertically developed, this largest of all clouds usually extends through all three cloud regions. Even the smallest cumulonimbus cloud dwarfs its neighbors in comparison. **Candidate 2**

Stratus clouds form when a large, moist air mass is gradually lifted over a wide area, typically due to gentle atmospheric processes such as warm air gliding over a cooler surface or radiative cooling near the ground. As the air rises slowly and cools, the moisture condenses into a uniform, flat cloud layer. Unlike cumulonimbus clouds, stratus clouds do not involve strong vertical convection and are usually associated with overcast skies, light drizzle, or mist rather than intense weather events. **Candidate 3**

Altocumulus (from Latin altus 'high' and cumulus 'heaped') is a middle-altitude cloud genus that belongs mainly to the stratocumuliform physical category, characterized by globular masses or rolls in layers or patches – the individual elements being larger and darker than those of cirrocumulus and smaller than those of stratocumulus. However, if the layers become tufted in appearance due to increased airmass instability, then the altocumulus clouds become more purely cumuliform in structure. Like other cumuliform and stratocumuliform clouds, altocumulus signifies convection. A sheet of partially conjoined altocumulus perlucidus is sometimes found preceding a weakening warm front, where the altostratus is starting to fragment, resulting in patches of altocumulus perlucidus between the areas of altostratus. Altocumulus is also commonly found between the warm and cold fronts in a depression, although this is often hidden by lower clouds. **Candidate 4**

**Output from Retrv-R1:**
<think> The user asks me to find a text that explains the formation mechanism of this type of cloud. To begin, I need to identify what type of cloud is shown in the image. From the image, we can observe that the cloud exhibits significant vertical development, stretching from the ground all the way to the top of the troposphere. Its top extends outward in an anvil shape, resembling a hammer or an iron anvil. The lower part of the cloud appears dense and dark, while the top is illuminated by sunlight, giving it a golden hue. These visual features strongly indicate that the cloud in the image is a cumulonimbus cloud. Therefore, my task is to locate a text that can explain the formation mechanism of cumulonimbus clouds. It is important that the text is specifically about cumulonimbus clouds—not any other type—and that it focuses primarily on the mechanism of formation, rather than on other aspects such as appearance or consequences. Let me examine the candidates. Candidate 3 and Candidate 4 are clearly not suitable, as their content pertains to other types of clouds. Now let me check Candidate 1 and Candidate 2. Candidate 1: <inspection-index-start>1<inspection-index-end>$T_{c_1}$. This text provides a detailed explanation of the entire process—from warm, moist air rising to the development of cumulonimbus clouds—which aligns well with the user's request. Although the final sentence of the text describes potential weather effects caused by cumulonimbus clouds, which falls outside the user's query, that part is minimal and occupies only a small portion of the text. Candidate 2: <inspection-index-start>2<inspection-index-end>$T_{c_2}$. This text mainly discusses the altitude and structural features of cumulonimbus clouds. While it briefly mentions their formation, that is not the main focus of the passage. Candidate 1 and Candidate 2 are not entirely focused on describing the formation mechanism of cumulonimbus clouds. Let me revisit Candidate 3 and Candidate 4—they are even less relevant to the query. Overall, since the explanation of the formation mechanism occupies the majority of Candidate 1, it is the most suitable retrieval result. </think> <answer>**1**</answer>

Figure 6: A qualitative example of the retrieval result generated from Retrv-R1.

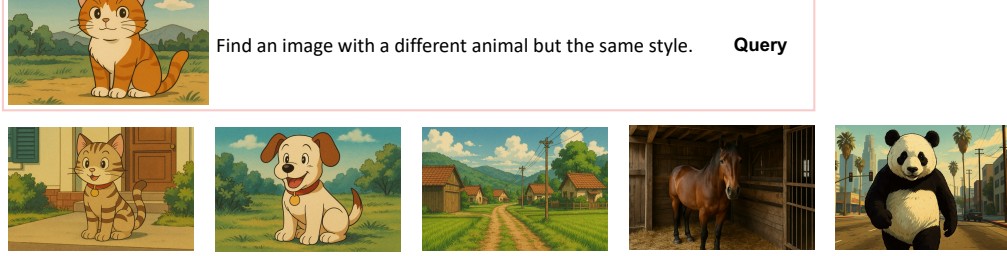

**Query**

Find an image with a different animal but the same style.

Candidate 1          Candidate 2          Candidate 3          Candidate 4          Candidate 5

**Output from Retrv-R1:**
<think> The user's request is to find an image with a different animal but in the same style as the input query image. To accomplish this task, I first need to identify the animal in the query image as well as the image's visual style. In the query image, a cat is sitting on a patch of land where golden and light green grass intersect. Therefore, the animal in the image is a cat. The image is rendered in a hand-drawn style with smooth color transitions, low contrast, and soft, low-saturation tones. The background resembles a watercolor picture book rather than a realistic setting, giving it the appearance of a retro Japanese animation style. Given this analysis, I need to find an image that does not contain a cat but features another kind of animal, and the style should also match the retro Japanese animation aesthetic. I will then analyze all the candidate images to identify the one that best meets these criteria. First, Candidate 1, Candidate 3, and Candidate 4 are clearly not the correct results. They either contain cats, do not include any animals, or have a visual style that significantly differs from the query image. Next, I turn to a deeper analysis of Candidate 2 and Candidate 5. First, Candidate 5 seems promising. It shares the cartoon-like style and with a different animal panda. Let me check it carefully: <inspection-index-start>5<inspection-index-end>$T_{c_5}$. I find that Candidate 5 includes urban elements such as city streets, tall buildings, and traffic lights, evoking a modern, urban atmosphere. In contrast, the query image depicts countryside elements such as open fields, grass, blue skies with white clouds, and rolling hills. Moreover, Candidate 5 features higher contrast and bolder colors, unlike the soft, low-saturation tones of the query image. Overall, Candidate 5 leans more toward an American comic style, whereas the query image is clearly in the retro Japanese anime tradition. Candidate 2, on the other hand, shares the same rural setting, soft tones, and low saturation as the query image. Its visual aesthetic is consistent with the retro Japanese animation style and contains a different animal dog, making it a better match. </think> <answer>**2**</answer>

Figure 7: A qualitative example of the retrieval result generated from Retrv-R1.

