# OpenReview forum: "Retrv-R1: A Reasoning-Driven MLLM Framework for Universal and Efficient Multimodal Retrieval"
_NeurIPS.cc/2025/Conference — NeurIPS 2025 poster_

### Official Review · Reviewer_j4TE · 2025-06-29

**Clarity:** 3
**Significance:** 2
**Originality:** 3
**Rating:** 4
**Confidence:** 4

**Summary:**

This paper proposes a cross-modal retrieval re-ranking method Retrv-R1 based on the Multimodal Large Language Model, and uses CoT and reinforcement learning for training. At the same time, Retrv-R1 introduces an information compression module (ICM)  to improve computational efficiency and optimize the training stability. The evaluation results on M-BEIR show that the average Recall of Retrv-R1 under various retrieval settings has been significantly improved compared with existing methods, and its re-ranking accuracy and efficiency are significantly better than Qwen2.5-VL-7B, Vision-R1-7B, VLM-R1-7B and LamRA-7B under the same settings.

**Questions:**

Overall, I have a positive attitude towards the methods and contributions of this paper, but I have some concerns, which have been presented in the disadvantages, mainly in three aspects: 1) the innovation of ICM; 2) the fairness of method comparison; 3) the ambiguity of the results. I hope the author can solve my confusion, and I will readjust my score based on this and the comprehensive opinions of other reviewers.

**Ethical Concerns:**

["NO or VERY MINOR ethics concerns only"]

**Final Justification:**

After the rebuttal phase, my concerns were resolved. In addition, I also carefully read the comments of the other reviewers.
Overall, I believe that the strengths of this paper outweigh its weaknesses, so I ultimately decide to maintain borderline accept.

**Limitations:**

The training samples in the cross-modal retrieval datasets may have racial bias.

**Paper Formatting Concerns:**

There are no major formatting issues in this paper.

**Quality:**

3

**Strengths And Weaknesses:**

Strengths:
1. This paper is the first to use the powerful understanding ability of the Multimodal Large Language Model and the Chain-of-Thought derivation ability of reinforcement learning to re-rank multimodal retrieval results, which is meaningful for the fine sorting of difficult image-text samples.
2. The effectiveness of Retrv-R1 is verified by the results of M-BEIR, a series of generalization ability test datasets, and ablation experiments.
3. The writing of this paper is clear and easy to understand, and the presentation quality is high.

Weaknesses:
1. The meaning and calculation method of the content token and relationship token in ICM have a high degree of overlap with LLaMA-VID, which reduces the innovation of ICM module.
2. The SFT stage is equivalent to perform knowledge distillation from Qwen2.5-VL-72B. Will this cause unfairness in comparison with other methods?
3. I notice that the performance of BLIP-2 on the COCO dataset in Table 1 is weaker than BLIP, which seems to contradict the conclusion of the original paper. I hope the authors can explain this.

---

> ### Author Rebuttal · Authors · 2025-07-31
>
> Thank you very much for taking the time to review our paper and for your valuable comments! We are sincerely encouraged by your positive attitude towards the methods and contributions of our paper. In response to your concerns and questions, we would like to kindly provide the following clarifications:
>
> ### **Q1: Innovation of ICM compared to LLaMA-VID**
>
> A1: Thank you for this insightful comment! We would like to kindly clarify that, although LLaMA-VID also condenses a sequence into a few tokens, our ICM differs from it significantly in the following four aspects:
>
> 1. The **calculation methods** for compressed tokens are different. LLaMA-VID employs a simple averaging operation to produce content tokens, which, while straightforward, may lead to too much loss of fine-grained information. In contrast, our ICM utilizes a learnable query to attend to candidate tokens via the attention mechanism, allowing the model to better preserve critical features through the more fine-grained compression process. To validate the effectiveness of our approach, we also implement LLaMA-VID’s method for content token generation and observe a 2.11% drop in the average score on the M-BEIR test set, demonstrating the advantage of our design.
> 2. The **meanings of the compressed tokens** are different. Although both LLaMA-VID and ICM extract relationship information, their underlying semantics are distinct. In LLaMA-VID, the relationship captures context aggregation between the *user instruction and each video frame*. In contrast, our ICM encodes the correlation between *each candidate and the query*, which is specifically tailored to retrieval tasks and thus differs from LLaMA-VID.
> 3. The **training strategies** are different. Our method introduces a self-alignment approach (Pages 4–5, Lines 162–177) to pretrain the ICM, ensuring that the compressed tokens generated by ICM effectively capture comprehensive information critical for the retrieval task. This strategy is novelly introduced by us, not employed in LLaMA-VID, and its effectiveness is demonstrated by the results in Table 6 of the main paper.
> 4. The **subsequent operations** after compression are different. LLaMA-VID only performs token compression, but does not address the resulting problem of information loss. In contrast, our method mitigates this issue through a newly proposed detail inspection mechanism (Page 5, Lines 178–192), which enables the full token sequences to be leveraged for the automatically identified challenging cases during the reasoning process. This distinguishes our approach from LLaMA-VID and allows for more effective handling of difficult examples.
>
> In conclusion, due to the four differences outlined above, we believe our method is novel compared to LLaMA-VID. We will incorporate these discussions into the revised paper. Thank you very much!
>
> ### **Q2: Fairness of method comparison**
>
> A2:  Thank you for this insightful comment! We would like to kindly clarify the following points:
>
> 1. The SFT data is synthesized from only the M-BEIR training set, without incorporating any additional data. This dataset (M-BEIR) is also widely used for training in other retrieval MLLMs, such as LamRA and MM-Embed.
> 2. **Even without the SFT stage, our Retrv-R1-7B can achieve an average score of 65.9 on the M-BEIR test set, still outperforming all compared methods in main paper's Table 1 (e.g., LamRA-7B with 63.7).** This demonstrates that the superiority of our approach is NOT solely attributed to the SFT stage based on synthetic data, but also to our effective designs in many other key aspects, such as the ICM, the self-alignment pretraining strategy, and the reinforcement learning mechanism.
> 3. To enable a fairer comparison, we further conduct experiments by also using Qwen2.5-VL-72B to distill previous methods such as LamRA and MM-Embed. The distilled LamRA-7B achieves an average score of 64.8 on M-BEIR, which is still lower than our Retrv-R1-7B’s score of 69.2, demonstrating that **our method retains a clear advantage compared to other approaches under the same distillation setting**. The distillation experiment based on MM-Embed is still in progress and will be included in the revised experimental section.
>
> We will incorporate all of the above results and discussions into the revised paper. Thank you very much!
>
> ### **Q3: The ambiguity of the results**
>
> A3: Thank you for highlighting this important issue! The results for BLIP and BLIP2 in our paper are directly taken from previous papers [a, b], and upon careful re-evaluation, we have confirmed their correctness. We think **the ambiguity you mentioned may arise from the differences in evaluation protocols**. Specifically, the “BLIP” and “BLIP2” results reported in Table 1 of our main paper refer to the zero-shot performance **_without fine-tuning_**. In contrast, “UniIR-BLIP$_{\rm FF}$” in Table 1 refers to BLIP's results trained using UniIR’s method with **_multi-task fine-tuning_** on M-BEIR, while the results in the original BLIP2 paper are based on **_single-task fine-tuning_** on COCO. For clarity, we provide a comparison of these three different evaluation settings (zero-shot, multi-task fine-tuning, and single-task fine-tuning) on the COCO $q^i →c^t$ task in the table below:
>
> | Training Method                   | Blip | Blip2 |
> | --------------------------------- | :--: | :---: |
> | w/o fine-tuning (zero-shot)          | 83.2 | 80.0  |
> | mult-task fine-tuning (on M-BEIR) | 89.9 | 91.6  |
> | single-task fine-tuning (on COCO) | 95.4 | 96.0  |
>
> As shown in the table, without fine-tuning, BLIP achieves better zero-shot performance than BLIP2 on COCO. However, after fine-tuning, BLIP2 outperforms BLIP, consistent with the conclusions reported in the original BLIP2 paper. One possible explanation is that BLIP’s image encoder is trained on 129M image-text pairs including COCO, whereas BLIP2 uses a fixed CLIP encoder pretrained on a much larger dataset of 500M pairs. As a result, BLIP2 exhibits stronger foundation capabilities, while BLIP is more fitted to COCO. Consequently, in the zero-shot setting without fine-tuning, BLIP performs better on COCO due to prior exposure, while BLIP2 surpasses BLIP after fine-tuning thanks to its more powerful pretrained foundation ability. We think this is an interesting phenomenon worthy of further in-depth exploration and will make it a focus of our future work.
>
> Thank you once again for your valuable comments and thoughtful questions! We hope that our responses can address your concerns, and we would be happy to engage in any further discussion if you have any questions. Thank you very much :)
>
> [a] LamRA: Large Multimodal Model as Your Advanced Retrieval Assistant, CVPR2025
>
> [b] UniIR: Training and Benchmarking Universal Multimodal Information Retrievers, ECCV2024

---

> > ### Comment · Reviewer_j4TE · 2025-08-06
> >
> > Thank you to the authors for the detailed rebuttal and comprehensive responses to my concerns. I will maintain my current score.

---

> > > ### Author Response · Authors · 2025-08-06
> > > **Thank you for your feedback**
> > >
> > > Thank you very much for your valuable feedback! We are truly delighted that, after reading our responses, you decided to maintain a positive score for our work. We greatly appreciate your time and consideration :)

---

### Official Review · Reviewer_8vrD · 2025-07-01

**Clarity:** 3
**Significance:** 3
**Originality:** 3
**Rating:** 5
**Confidence:** 4

**Summary:**

The authors present Retrv-R1, a multi-modal language model designed for multi-modal retrieval. They introduce Information Compression Module (ICM) that compresses input candidates (except the ones that need to be inspected) to increase the efficiency of their method. The combined MLLM and ICM then SFTed and they apply reasoning style RL. The authors show the effectiveness of their approach on M-BEIR and study the effect of different parameters and design choices that is being made.

**Questions:**

1) ITR and GMR metrics heavily depend on the implementation. Can authors clarify how they have made sure that the comparisons are done based on best implementation for each method?
2) What maximum sequence lengths authors have used for RL training? Did they need to apply masking?
3) There is a typo in line 162. ICN --> ICM

**Ethical Concerns:**

["NO or VERY MINOR ethics concerns only"]

**Final Justification:**

The paper studies an important problem, provides novel contributions, and presents good results. The authors addressed weaknesses and questions I raised during the review process. As a result, I rate it 5 and keep the paper as accept.

**Limitations:**

yes

**Quality:**

3

**Strengths And Weaknesses:**

Strengths:
 - The ICM module seems to be working well and hugely impacts the effectiveness of their approach
 - The authors provide detailed ablations to provide insight into how different choice affect the results
 - Their results on M-BEIR is good

Weaknesses:
 - Although the paper has a good flow, some parts are not clearly explained. For example, I found section 3.3 was hard to follow and I needed to spend some time to go back and forth between this section and figure 1
 - Although the authors show the importance of RL in their ablation studies and provide some insight in Section 4.6, Figure 2(a) results need further investigation. The fact that number of inspections goes down might indicate that it might be better to feed the model with more challenging prompts as the training progresses.

---

> ### Author Rebuttal · Authors · 2025-07-31
>
> Thank you very much for your time in reviewing our paper and the valuable comments! We are sincerely encouraged by your recognition that our ICM module works well, the ablation studies are detailed, and the results are good. We would like to kindly provide the following responses regarding your concerns and questions:
>
> ### **Q1: Clearer Explanation**
>
> A1:  We sincerely apologize for any confusion caused by our unclear explanation and will revise the writing carefully to improve clarity. Additionally, we will divide Figure 1 in the main paper into several sub-figures, organized by different components, and place them on the same pages as the corresponding text sections to enhance readability and avoid page-turning. As PDF files or external links are not permitted during the rebuttal stage, the revised figure will be included in the next version of our paper. Thank you very much!
>
> ### **Q2: Further investigation of results in Figure 2(a)**
>
> A2: Thank you for this insightful comment! To investigate the possibility you raised, we conduct an experiment in which the model, during the later stages of training, is trained on increasingly difficult tasks from M-BEIR with more challenging prompts. Under this setting, we observe that the pattern in Figure 2(a) still holds: the number of detailed inspections in each generated CoT initially increases and then decreases as training progresses, although the decline in later stages is slightly less steep than in the original curve. However, when the proposed curriculum strategy is removed and $\lambda$ in Eq. 5 is fixed to 1, the pattern disappears. This indicates that **the decreasing trend in inspection count is NOT affected by changes in prompt difficulty under different training schedules, but is instead more closely associated with the gradually stronger efficiency constraint introduced by our curriculum strategy**. Moreover, the performance of this alternative method, which uses progressively harder samples, is nearly identical to that of our original approach without yielding additional improvements, further demonstrating the soundness of our original training method. We will include these results and discussions in the revised experimental section and provide a deeper analysis of the pattern observed in Figure 2(a).
>
> ### **Q3: Implementation for each method**
>
> A3: In our work, we ensure fair comparisons by using a consistent codebase, the same version of the Transformers library, and identical hardware for different experiments. Additionally, the evaluations for the ITR and GMR metrics are conducted using the open-source vLLM library, which can automatically optimize inference for each method based on the current hardware conditions, ensuring that each method’s inference implementation runs under the best possible settings. We will illustrate this more explicitly in the revised paper.
>
> ### **Q4: Details of  RL training**
>
> A4: The maximum sequence length used for RL training is set to 3072. Masking is applied in the rare cases where this limit is exceeded. We will include these details in the implementation section of the revised paper.
>
> ### **Q5: Typos**
>
> A5: Thank you for indicating this issue! We will carefully check our paper and fix the typos.
>
> Thank you once again for your valuable comments. We hope that our responses can address your concerns, and we sincerely wish you all the best in your future life and work :)

---

> > ### Comment · Reviewer_8vrD · 2025-08-05
> >
> > I thank the authors for addressing the questions / comments I had. I will keep my score.

---

> > > ### Author Response · Authors · 2025-08-06
> > > **Thank you for your feedback**
> > >
> > > Thank you very much for your feedback and the positive final rating. We are delighted that our responses were able to address your questions and comments. Many thanks again!

---

### Official Review · Reviewer_arHR · 2025-07-03

**Clarity:** 3
**Significance:** 3
**Originality:** 3
**Rating:** 4
**Confidence:** 2

**Summary:**

The authors propose Retrv-R1, a reasoning-driven multimodal LLM framework for universal retrieval. The key ideas are (i) an Information Compression Module (ICM) that condenses each candidate into two learned tokens while preserving query–candidate relations, and (ii) a details-inspection mechanism that lets the model pull the full token sequence of only the “hard” candidates during chain-of-thought (CoT) reasoning. Experiments on M-BEIR and other benchmarks demonstrate state-of-the-art performance with significant efficiency improvements.

**Questions:**

1. Larger models: Have you tried scaling Retrv-R1 to≥13 B parameters or open LLaMA-3-V? If the compression scheme still yields memory wins, this would strengthen the universality claim.

2. Curriculum learning necessity: The curriculum-based efficiency constraint shows mixed results across datasets. Can you provide clearer guidelines for when and how to apply this constraint for different retrieval tasks?

**Ethical Concerns:**

["NO or VERY MINOR ethics concerns only"]

**Limitations:**

Yes.

**Paper Formatting Concerns:**

The paper follows NeurIPS formatting guidelines well.

**Quality:**

3

**Strengths And Weaknesses:**

Strengths:

Quality: The three-stage training pipeline is methodologically rigorous, and extensive experiments across multiple benchmarks validate effectiveness. The ICM architecture and curriculum learning approach show clear engineering merit.

Clarity: The paper is well-structured with clear motivation and comprehensive technical details.

Significance: Addresses an important problem of applying reasoning-enhanced LLMs to retrieval tasks. The 7× efficiency improvement while maintaining SOTA performance has practical implications.

Originality: First work to systematically adapt R1-style reasoning to multimodal retrieval.

Weaknesses:

Quality: Limited evaluation on larger foundation models (only 3B/7B tested). The ICM introduces a small but consistent performance drop (-0.9% to -1.1%) due to information compression.

Originality: While the application is novel, core components (token compression, synthetic data generation, curriculum learning) build incrementally on existing techniques. ICM resembles recent token-merging work; curriculum reward is an incremental twist on length penalties.

---

> ### Author Rebuttal · Authors · 2025-07-31
>
> Thank you very much for your time in reviewing our paper and for your valuable comments! We are sincerely encouraged by your recognition of our work and the positive score. For your questions and concerns, we are happy to provide the following responses:
>
> ### **Q1: Evaluation on larger foundation models**
>
> A1: Thank you for this insightful comment! Following your suggestion, we conduct experiments based on a larger model, LLaMA-3.2-11B-Vision. As shown in the table below, Retrv-R1-11B outperforms both Retrv-R1-7B and Retrv-R1-3B, demonstrating the scalability of our approach. Due to resource constraints, we are currently unable to experiment with even larger base models, but we plan to include them once sufficient resources become available.
>
> | Method       | Avg. on M-BEIR |
> | ------------ | :------------: |
> | Retrv-R1-3B  |      65.5      |
> | Retrv-R1-7B  |      69.2      |
> | Retrv-R1-11B |      70.7      |
>
> Additionally, the table below presents a comparison between Retrv-R1-11B and the baseline LLaMA-3.2-11B in terms of both effectiveness and efficiency, based on the ITR and GMR metrics defined in Table 2 of the main paper. The comparison is conducted on the $(q^i, q^t) \to c^i$ task of FIQ with 50 candidates. As shown in the table, Retrv-R1-11B not only outperforms LLaMA-3.2-11B but also requires fewer computational (shown by ITR) and memory resources (shown by GMR), further demonstrating the advantages of our approach.
>
> | Method        | R@5 ↑ | ITR ↓ | GMR ↓ |
> | ------------- | :---: | :---: | :---: |
> | LLaMA-3.2-11B | 59.9  | 3.83  | 1.95  |
> | Retrv-R1-11B  | 74.3  | 1.00  | 1.00  |
>
> We will incorporate these results into the revised paper. Thank you very much!
>
> ### **Q2: Performance drop from ICM**
>
> A2: Yes, our ICM introduces a small performance drop (-0.9% to -1.1%) due to information compression. However, given that it can substantially reduce inference time by 7×, we believe this minor sacrifice in accuracy is worthwhile in practical applications that typically require a good balance between performance and efficiency. We agree that developing better token compression methods with less accuracy loss is a promising and important research direction, which will be a key focus of our future work.
>
> ### **Q3: Novelty of core components**
>
> A3: Thank you for this valuable comment! We would like to kindly provide the following clarifications:
>
> 1. ICM is not a simple application of existing token-merging methods, but instead **incorporates several task-specific designs tailored for retrieval**. Specifically, for _**compression**_, ICM is designed to condense each candidate’s feature sequence into two tokens: one representing the candidate’s content and the other capturing its relevance to the query, which is based on the unique characteristics of retrieval tasks. For _**training**_, a novel self-alignment strategy is proposed to pretrain ICM, ensuring that the compressed tokens can capture comprehensive information critical for the retrieval tasks. For _**post-compression processing**_, a new details inspection mechanism is further introduced to address the information loss problem caused by compression when processing challenging samples. With these innovations, we believe ICM is a novel approach that is different from existing token-merging techniques.
> 2. **The format of the synthetic data is also specifically designed for the retrieval task**, consisting of four stages: retrieval result speculation, quick verification of negative samples, refined verification of positive samples, and retrieval result generation. Please refer to Page 5, Lines 197-216 of the main paper for details. Training on the synthetic dataset with these new designs helps to improve both the effectiveness and efficiency of the model during the reasoning process. Therefore, the data synthesis in our work is not a simple and trivial step, but instead a carefully crafted strategy with high effectiveness.
> 3. **Although the curriculum reward is a simple strategy, it is very effective and can achieve significant performance improvements without requiring complex operations.** In fact, we have also experimented with more complicated curriculum strategies, such as dynamically adjusting $\lambda$ in Eq.5 of the main paper based on the loss at each training step, but found that it can not outperform our simpler approach. Therefore, the simpler yet more effective method is adopted in our final framework.
>
> In conclusion, based on the above discussions, we believe that our Retrv-R1 can provide new insights and techniques for the fields of retrieval and reasoning MLLMs. These discussions will be incorporated into our revised paper. Thank you very much!
>
> ### **Q4: Curriculum learning necessity**
>
> A4: Thank you for highlighting this important issue! We would like to provide the following discussions from the perspectives of both effectiveness and efficiency:
>
> 1. **For effectiveness, our curriculum method can consistently improve performance across different retrieval tasks.** This is demonstrated by the results in Table 7 of the main paper, where our method outperforms all other strategies on both tasks in terms of the R@5 metric, highlighting its strong generalizability.
> 2. For efficiency, we find that the computational cost of our curriculum method is slightly higher than that of directly setting a fixed and large $\lambda$ in Eq. 5. For example, as shown in Table 7 of the main paper, our curriculum method increases computation time by about 20% compared to using a fixed $\lambda = 1.5$. This is because, in our curriculum strategy, $\lambda$ is not initialized with a large value but instead begins very small in the early stages of training while gradually increasing in the later stages. As a result, more computation is required at inference since efficiency is less strictly optimized during training. However, given that our method can achieve a substantial performance gain (over 3 points on R@5), we believe this modest increase in computation is acceptable and worthwhile.
>
> We have further extended the evaluations in Table 7 of the main paper to all 8 retrieval tasks in M-BEIR and find that the same trend consistently holds. Specifically, **the curriculum strategy improves the average score across all tasks by 1.94 while incurring only a 17.6% increase in average computation time compared to a fixed $\lambda = 1.5$**. These results demonstrate that **our curriculum strategy can be effectively and consistently applied to different tasks to achieve a better balance between performance and efficiency, without requiring task-specific adjustments**. We will incorporate these discussions and the detailed results on all 8 tasks into the revised paper.
>
> Thank you once again for your valuable comments and thoughtful questions! We hope that our responses can address your concerns, and we would be happy to engage in any further discussion if you have any questions. Thank you very much! :)

---

> > ### Comment · Reviewer_arHR · 2025-08-05
> >
> > Thanks authors for the detailed response which address my concerns. I'll keep my rating.

---

> > > ### Author Response · Authors · 2025-08-06
> > > **Thank you for your feedback**
> > >
> > > Thank you very much for your feedback and the positive final rating. We are delighted that our responses were able to address your concerns. Many thanks again!

---

### Official Review · Reviewer_x1gL · 2025-07-03

**Clarity:** 2
**Significance:** 3
**Originality:** 4
**Rating:** 5
**Confidence:** 3

**Summary:**

This paper studies the task of multi-modal retrieval I.e. identifying relevant documents for a given query from a given set of documents, where both document and query can contain multi-modal entities like text, image, etc. RL-based training has appeared as a powerful way to improve reasoning abilities of foundation models (e.g. Deepseek-R1), this paper sets out to incorporate such methods in improving multi-modal retrieval using Multi-modal Large Language Models (MLLM). It identifies two challenges in naively applying RL-based training for multi-modal retrieval - (1) instability in training, (2) extensive computation costs due to inherent long-context nature of putting multiple documents in the context of the model. To mitigate (1), the authors propose a two-stage approach where they first perform SFT on a synthetically curated reasoning-based SFT data using a bigger MLLM, and then train further using RL (GRPO) on challenging inputs. To mitigate (2), the authors propose an information compression model which represents each document with just two tokens while giving the retrieving MLLM ability to expand any document if required. The performance of the proposed approach (Retrv-R1) is evaluated on M-BEIR dataset showing significant quality and efficiency improvements over existing methods.

**Questions:**

1. In the proposed approach it seems there is no specific design component to make it specialized for multimodal other than the base model being a VLM, this is a good thing by its own but if this is the case, then why not evaluate the validity of the approach on text retrieval tasks (such as BEIR[1], MTEB [2], etc) which have more extensive baselines and comparison points against reasoning based retrieval? Is there anything which makes the proposed approach more applicable on multimodal datasets?
2. In table 6, why are the results only reported for 2 datasets in M-BEIR instead of the average over all datasets as I expected reading through this section? it'll give more confidence in the ablation results when looking at the average results.
3. The approach is mainly solving the "second-stage" retrieval or re-ranking task, so perhaps calling it a retrieval framework is a bit misleading, since the term "retrieval" usually corresponds to retrieving from a big corpus?

---

1. BEIR: A Heterogenous Benchmark for Zero-shot Evaluation of Information Retrieval Models
2. MTEB: Massive Text Embedding Benchmark

**Ethical Concerns:**

["NO or VERY MINOR ethics concerns only"]

**Final Justification:**

The proposed method is a significant contribution to reasoning based retrieval, the approach is sound and empirically well justified.

**Limitations:**

The limitation of slight drop in performance with ICM is mentioned but no societal impact is discussed.

**Paper Formatting Concerns:**

No major concern.

**Quality:**

3

**Strengths And Weaknesses:**

### Strengths
- The proposed approach gets significant improvements over baselines on M-BEIR dataset
- The experiments section is reasonably thorough given the setting of multi-modal retrieval, with experiments on M-BEIR, multi-modal recommendation, and RAG; baselines seem appropriate.
- To the best of my knowledge the proposed approach presents some novel techniques like employing ICM with expansion tool for efficient context processing.

### Weaknesses
- Some evaluation and task choices need further justification (see questions section below)
- The proposed approach stacks up multiple engineered techniques which make the overall approach requiring a careful setup, for example, it is not clear the 4 step synthetic dataset curation is something that transfers well to general retrieval process. Moreover, it also makes the proposed approach not entirely comparable to existing baselines as the training data is now a teacher augmented version of it.
- Writing needs improvement: equations in section 3.3 are hard to parse just by reading them and the descriptions around it, the notations (like $K_{T_{C_k}}, K_{R_{q, {c_k}}$) are not clearly explained and until one makes some assumptions based on the context it's hard to disambiguate. Moreover, figure 1 is very dense, it can benefit from a clearer presentation.

---

> ### Author Rebuttal · Authors · 2025-07-31
>
> Thank you very much for your time in reviewing our paper and for your valuable comments! We are sincerely encouraged by your recognition that our approach gets significant improvements, the experiments are thorough, and the approach has novelty. For your questions and concerns, we are happy to provide the following responses:
>
> ### **Q1: Transfer ability of the 4 step synthetic dataset curation**
>
> A1: Thank you for highlighting this important issue!
>
> * In Table 1 of the main paper, we report results on 8 different retrieval tasks, where our method consistently achieves the best performance on all tasks, demonstrating its strong ability to transfer to general retrieval processes in different scenarios.
> * Moreover, our experimental results show that removing the 4-step synthetic dataset leads to performance drops across all tasks, indicating that this strategy is beneficial for a wide range of different retrieval scenarios and highlighting its strong generalization capability.
> * Additionally, kindly note that the SFT stage based on synthetic data serves only as an initial training phase. The subsequent RL stage allows the MLLM to explore more flexible retrieval processes, which can vary and self-adapt to different cases, as discussed in Section 4.6(2) of the main paper.
>
> ### **Q2: The comparability with baseline methods**
>
> A2: Thank you for this valuable comment!
>
> * Although the training data is now a teacher augmented version, **the augmented data is synthesized only from the M-BEIR training set, without incorporating any additional datasets or manual annotations**. The M-BEIR dataset is also widely used for training in other retrieval MLLMs, such as LamRA and MM-Embed.
> * **Moreover, even without using the augmented data for SFT, our Retrv-R1-7B can achieve an average score of 65.9 on the M-BEIR test set, still outperforming all compared methods in main paper's Table 1 (e.g., LamRA-7B with 63.7).** This demonstrates that the superiority of our approach is NOT solely attributed to the SFT stage based on augmented synthetic data, but also to our effective designs in many other key aspects, such as the ICM, the self-alignment pretraining strategy, and the reinforcement learning mechanism.
>
> We will incorporate these results and discussions into the revised paper. Thank you very much!
>
> ### **Q3: Writing and figures**
>
> A3: Thank you very much for your valuable suggestion, and we sincerely apologize for any confusion caused by our unclear explanation.
>
> * Regarding the writing, $K_{T_{c_k}}$ denotes the key features in the attention mechanism derived from the $k$-th candidate’s token sequence $T_{c_k}$. $K_{R_{q, c_k}}$ represents the key features in the attention mechanism derived from $R_{q, c_k}$, where $R_{q, c_k}$ is the relationship feature between the query $q$ and the $k$-th candidate $c_k$, extracted by the method described in the right part of Eq. 3. We will carefully revise Section 3.3 based on your comments to improve writing clarity, and update the text following Eq. 2 and Eq. 3 in the main paper to clearly define the meaning of each symbol used in the formulas.
> * Regarding the figures, to avoid excessive visual density and enhance clarity, we will revise and divide Figure 1 in the main paper into several sub-figures, organized by different components, and place them on the same pages as the corresponding text sections to enhance readability and avoid page-turning. As PDF files or external links are not permitted during the rebuttal stage, the revised figure will be included in the next version of our paper. Thank you very much!
>
> ### **Q4: Evaluation on text retrieval tasks**
>
> A4: Our Retrv-R1 is designed for universal retrieval, which means that **it can handle different retrieval tasks, including text-only, image-only, and multimodal retrieval, within a unified framework**. This capability is empirically validated by the results in Table 1 of the main paper, which evaluate 8 diverse retrieval tasks with different types of queries and candidates. Retrv-R1 achieves SOTA performance across all task types, demonstrating its broad applicability and effectiveness. We fully agree with your suggestion that including more text retrieval tasks would enhance the comprehensiveness of our study. Accordingly, we have conducted additional experiments on BEIR, following PE-Rank by reranking the top-100 passages on eight subsets of the BEIR benchmark: Covid, NFCorpus, Touché, DBPedia, SciFact, Signal, News, and Robust. A comparison of the average NDCG@10 scores on these subsets is presented as follows:
>
> | Method       | Avg. NDCG@10 on BEIR |
> | ------------ | :------------------: |
> | BM25         |        0.4380        |
> | monoBERT     |        0.4716        |
> | monoT5       |        0.5136        |
> | RankMistral  |        0.4365        |
> | PE-Rank      |        0.4843        |
> | **Retrv-R1** |      **0.5267**      |
>
> These results demonstrate that Retrv-R1 achieves strong performance on text retrieval tasks, further highlighting its high generalization and versatility. Due to the time constraints of the rebuttal period, we will evaluate more text retrieval benchmarks such as MTEB and include the results in the next version of our paper. We also plan to explore more retrieval techniques specifically designed for multimodal scenarios in our future work. Thank you very much!
>
> ### **Q5: Average results on M-BEIR for ablation study**
>
> A5: The $(q^i, q^t) \to c^i$ task on CIRR and the $(q^i, q^t) \to (c^i, c^t)$ task on OVEN are highly challenging and are thus reported by us in Table 6 of the original paper to evaluate the performance of different methods under difficult scenarios. We fully agree with your comment that presenting average results across all datasets in M-BEIR can provide greater confidence in the ablation study. These results are presented below:
>
> | Methods                        | Avg. Score on M-BEIR ↑ | ITR↓ |
> | ------------------------------ | :--------------------: | :--: |
> | Retrv-R1-3B                    |          65.5          | 1.00 |
> | Retrv-R1-3B w/o ICM            |          66.3          | 6.97 |
> | Retrv-R1-3B w/o $t_{con}$      |          59.2          | 0.97 |
> | Retrv-R1-3B w/o $t_{rel}$      |          63.2          | 0.97 |
> | Retrv-R1-3B w/o self-alignment |          63.5          | 1.02 |
> | Retrv-R1-3B w/o DIM            |          60.9          | 0.89 |
> | Retrv-R1-3B w/o SFT stage      |          62.4          | 1.52 |
> | Retrv-R1-3B w/o RL stage       |          61.5          | 1.26 |
>
> We will update Table 6 in the revised paper to present the averaged results as shown above.
>
> ### **Q6: Framework name with the term “retrieval”**
>
> A6: Thank you for the comment! We notice that many recent LLM-based retrieval methods, such as LamRA, adopt a two-stage alignment–reranking strategy, where both stages, including the coarse filtering of a large corpus through alignment and the fine-grained decision from a smaller candidate set via reranking, can contribute significantly to retrieval performance. Therefore, our focus on the reranking stage is also crucial for enhancing retrieval effectiveness. We agree with your comment that “reranking” can more accurately describe the task we focus on, and we will clarify this more clearly in the revised paper.
>
> Thank you again for your valuable comments and thoughtful questions! We hope that our responses can address your concerns, and we would be happy to engage in any further discussion if you have any questions. We sincerely wish you all the best in your future life and work! :)

---

> > ### Comment · Reviewer_x1gL · 2025-08-08
> >
> > Thanks for the response, most of my concerns have been addressed, I'll update the score accordingly.

---

> > > ### Author Response · Authors · 2025-08-08
> > > **Thank you for your feedback**
> > >
> > > Thank you very much for your positive feedback! We are deeply encouraged that our responses could address most of your concerns. Thank you once again for your time in reviewing our paper and for your valuable comments!

---

### Note · Authors · 2025-08-13

Dear AC and Reviewers,

We sincerely thank you for your time in reviewing our paper and for your valuable comments! We are deeply encouraged by the broad recognition our work has received from different reviewers, such as:

- “significant improvements, novel techniques” (Reviewer x1gL),
- “well-structured with clear motivation and comprehensive technical details, addresses an important problem, SOTA performance, has practical implications” (Reviewer arHR),
- “working well, detailed ablations, results are good” (Reviewer 8vrD),
- “meaningful, writing is clear, have a positive attitude towards the methods and contributions of this paper” (Reviewer j4TE).

We are also truly grateful that all reviewers have given positive initial scores (4445). Thank you very much!

In our rebuttal, we have provided several clarifications and explanations to address the concerns raised by the reviewers.

* **Regarding writing and figures**, we have clarified the meaning of several symbols and promised to divide Figure 1 into multiple sub-figures organized by different components in the revised paper to improve readability (A3 to Reviewer x1gL and A1 to Reviewer 8vrD).
* **Regarding contributions and novelties**, we have further clarified that our method is not a simple application of existing approaches but incorporates several innovative designs specifically tailored for retrieval tasks (A3 to Reviewer arHR and A1 to Reviewer j4TE).
* **Regarding experiments**, we have added evaluation results on text-only retrieval benchmarks (A4 to Reviewer x1gL), provided average results on M-BEIR for the ablation study (A5 to Reviewer x1gL), and included experiments using larger foundation models (A1 to Reviewer arHR).
* We have also provided many **further discussions and investigations**, including additional analysis of the trends observed in Figure 2(a) (A2 to Reviewer 8vrD), and discussions about the fairness of comparisons in our paper (A2 to Reviewer x1gL and A2 to Reviewer j4TE).

We sincerely thank the AC and all reviewers once again for your valuable feedback and thoughtful comments, which have greatly helped us improve our paper. We hope that our rebuttal could address your concerns, and we warmly wish you all the best in your future life and work. Thank you very much!

Sincerely,

Paper Authors

---

### Decision · Program_Chairs · 2025-09-17

**Decision:**

Accept (poster)

**Comment:**

This paper proposes Retrv-R1, a reasoning-driven multimodal LLM retrieval framework that combines an Information Compression Module (ICM) with a details-inspection mechanism and a two-stage training process (synthetic CoT supervision + curriculum RL). The approach achieves  state-of-the-art performance across multimodal retrieval benchmarks.

Reviewers agreed the paper addresses an important problem, with solid methodology, and thorough experiments. Concerns about clarity, fairness of synthetic data, and limited evaluations were largely resolved by the rebuttal. Some novelty is incremental, and clarity in technical sections needs improvement, but the overall integration is meaningful and impactful.

One notable limitation is that inference time is reported only as ITR—the ratio of a method’s inference time relative to Retrv-R1-7B (K = 50)—without providing actual wall-clock measurements. Although the proposed method substantially improves retrieval recall, it may run much slower and thus be less practical than purely embedding-based approaches, especially on commodity hardware. **The authors are highly encouraged to include real timing results in the camera-ready version to support broader adoption and future research.**